# Status and physiological significance of circulating adiponectin in the very old and centenarians: an observational study

Takashi Sasaki[1]*, Yoshinori Nishimoto[1,2], Takumi Hirata[1,3], Yukiko Abe[1], Nobuyoshi Hirose[4], Michiyo Takayama[1,5], Toru Takebayashi[6], Hideyuki Okano[7], Yasumichi Arai[8]

[1]Center for Supercentenarian Medical Research, Keio University School of Medicine, Tokyo, Japan; [2]Department of Neurology, Keio University, School of Medicine, Tokyo, Japan; [3]Institute for Clinical and Translational Science, Nara Medical University, Nara, Japan; [4]Houtokukai Utsunomiya Hospital, Tochigi, Japan; [5]Center for Preventive Medicine, Keio University School of Medicine, Tokyo, Japan; [6]Department of Preventive Medicine and Public Health, Keio University School of Medicine, Tokyo, Japan; [7]Department of Physiology, Keio University School of Medicine, Tokyo, Japan; [8]Faculty of Nursing and Medical Care, Keio University, Tokyo, Japan

## Abstract

**Background:** High levels of circulating adiponectin are associated with increased insulin sensitivity, low prevalence of diabetes, and low body mass index (BMI); however, high levels of circulating adiponectin are also associated with increased mortality in the 60–70 age group. In this study, we aimed to clarify factors associated with circulating high-molecular-weight (cHMW) adiponectin levels and their association with mortality in the very old (85–89 years of age) and centenarians.

**Methods:** The study included 812 (women: 84.4%) for centenarians and 1498 (women: 51.7%) for the very old. The genomic DNA sequence data were obtained by whole-genome sequencing or DNA microarray-imputation methods. Least Absolute Shrinkage and Selection Operator (LASSO) and multivariate regression analyses were used to evaluate cHMW adiponectin characteristics and associated factors. All-cause mortality was analyzed in three quantile groups of cHMW adiponectin levels using Cox regression.

**Results:** The cHMW adiponectin levels were increased significantly beyond 100 years of age, were negatively associated with diabetes prevalence, and were associated with SNVs in *CDH13* (p=2.21 × 10$^{-22}$) and *ADIPOQ* (p=5.72 × 10$^{-7}$). Multivariate regression analysis revealed that genetic variants, BMI, and high-density lipoprotein cholesterol (HDLC) were the main factors associated with cHMW adiponectin levels in the very old, whereas the BMI showed no association in centenarians. The hazard ratios for all-cause mortality in the intermediate and high cHMW adiponectin groups in very old men were significantly higher rather than those for all-cause mortality in the low-level cHMW adiponectin group, even after adjustment with BMI. In contrast, the hazard ratios for all-cause mortality were significantly higher for high cHMW adiponectin groups in very old women, but were not significant after adjustment with BMI.

**Conclusions:** cHMW adiponectin levels increased with age until centenarians, and the contribution of known major factors associated with cHMW adiponectin levels, including BMI and HDLC, varies with age, suggesting that its physiological significance also varies with age in the oldest old.

**Funding:** This study was supported by grants from the Ministry of Health, Welfare, and Labour for the Scientific Research Projects for Longevity; a Grant-in-Aid for Scientific Research (No 21590775, 24590898, 15KT0009, 18H03055, 20K20409, 20K07792, 23H03337) from the Japan Society for

*For correspondence:
sasasa@z5.keio.jp

the Promotion of Science; Keio University Global Research Institute (KGRI), Kanagawa Institute of Industrial Science and Technology (KISTEC), Japan Science and Technology Agency (JST) Research Complex Program "Tonomachi Research Complex" Wellbeing Research Campus: Creating new values through technological and social innovation (JP15667051), the Program for an Integrated Database of Clinical and Genomic Information from the Japan Agency for Medical Research and Development (No. 16kk0205009h001, 17jm0210051h0001, 19dk0207045h0001); the medical-welfare-food-agriculture collaborative consortium project from the Japan Ministry of Agriculture, Forestry, and Fisheries; and the Biobank Japan Program from the Ministry of Education, Culture, Sports, and Technology

## Editor's evaluation

This study provides useful descriptive findings regarding levels of adiponectin and its associations with genetic variants and age in very old and centenarian men and women. Due to some missing data, the evidence supporting the conclusions about cross-sectional determinants of circulating levels of adiponectin in centenarians is incomplete. The evidence for conclusions about relationships between adiponectin and survival is also incomplete because of some missing data and unadjusted survival analyses.

## Introduction

Adiponectin is an adipocyte-derived hormone that plays a vital role in metabolism, including lipid and glucose metabolism, and occurs in circulation at concentrations of up to 0.05% of total plasma protein (*Scherer et al., 1995*; *Ye and Scherer, 2013*). Circulating adiponectin forms three major multimer complexes, including a trimer, hexamer, and high-molecular-weight form. Among these forms, circulating high-molecular-weight (cHMW) adiponectin shows more potent biological activity than that of the other two forms (*Pajvani et al., 2003*). Previous studies in the mouse model studies have shown that cHMW adiponectin enhances insulin sensitivity and plasma lipid clearance; high levels of cHMW adiponectin improve the stability of lipid homeostasis and provided systemic tolerance to obesity under normal physiological conditions (*Combs et al., 2004*; *Qiao et al., 2008*; *Asterholm and Scherer, 2010*). Adiponectin knock-out mice showed mild or moderate insulin resistance, which is exacerbated by a high-fat diet (*Kubota et al., 2002*; *Maeda et al., 2002*). However, adiponectin knock-out mice are viable under regular physiological conditions, indicating that adiponectin is not essential for survival under regular dietary conditions (*Kubota et al., 2002*; *Maeda et al., 2002*). Therefore, adiponectin function is considered inconspicuous under normal conditions and should become prominent under physiological stress such as hyperglycemia.

In humans, adiponectin shows strong negative associations with body mass index (BMI), the prevalence of type 2 diabetes mellitus (T2DM), and hypertension (*Li et al., 2009*; *Arita et al., 1999*; *Adamczak et al., 2003*). However, high levels of adiponectin have also been associated with an increased risk of cardiovascular disease (CVD) in adults in their 60s and 70s (*Kizer et al., 2008*; *Poehls et al., 2009*; *Kim-Mitsuyama et al., 2019*; *Choi et al., 2015*). These contradictory findings indicate that environmental and related physiological changes could alter the level and function of adiponectin; therefore, the analysis of adiponectin in adults aged 80 years and older would be essential to elucidate the significance of adiponectin in aging.

Centenarians are individuals aged 100 years and older and characterized by a low incidence of life-threatening diseases, such as CVD and T2DM. They serve as potential models for successful aging (*Davey et al., 2012*; *Arai et al., 2015*). Previous studies have reported that cHMW adiponectin levels increase with age and, specifically, that centenarians show comparatively high levels (*Arai et al., 2019*; *Cnop et al., 2003*). Low BMI may contribute to high adiponectin levels and insulin resistance in older adults ages above 60 years, and transgenic mouse models have shown prolonged health span and median lifespan. However, the physiological significance of high cHMW adiponectin levels in adults aged above 80 years is still unclear (*Muratsu et al., 2021*; *Li et al., 2021*; *Cohen et al., 2022*). To provide evidence for understanding the physiological function and significance of adiponectin in the oldest old, this study aimed to determine the status and factors associated with cHMW adiponectin levels in 2310 adults aged ≥85 years, including 812 centenarians.

## Methods

### Study populations

We used data from four prospective cohort studies of the oldest old in Japan: the Tokyo Centenarian Study (TCS) and Japanese Semi-supercentenarian Study (JSS) for centenarians and the Tokyo Oldest Old Survey on Total Health (TOOTH) and Kawasaki Aging Wellbeing Project (KAWP) for the very old (aged 85–99 years). Recruitment was conducted as previously described (*Arai et al., 2015*; *Arai et al., 2014*; *Gondo et al., 2006*; *Takayama et al., 2007*; *Arai et al., 2010*; *Arai et al., 2021*; *Sasaki et al., 2021b*). From the TCS and JSS, 155 participants were excluded due to a lack of cHMW adiponectin level data; thus, 812 centenarians were enrolled (127 men and 685 women with a median age of 105.3 [interquartile range (IQR): 100.9–106.8] and 106.0 years [IQR: 103.9–107.2], respectively). The TOOTH and KAWP surveys are community-based prospective cohort studies of individuals between 85 and 102 years (TOOTH) and 85 and 90 years (KAWP), respectively. Data for 542 (236 men and 306 women) and 1026 (513 men and 513 women) individual medical examinations are included in the TOOTH and KAWP studies, respectively. Of these, 63 individuals from the TOOTH study were excluded because they were older than 90, and 7 individuals from the KAWP study were excluded due to a lack of cHMW adiponectin level data; thus, 1498 individuals were enrolled as the very old (724 men and 774 women with a median age of 86.9 [IQR: 85.9–88.2] and 87.0 years [IQR: 86.0–88.4], respectively, *Figure 1—figure supplements 1, 2*, *Supplementary file 1*).

All of the KAWP, TOOTH, TCS, and JSS have been managed by the Center for Supercentenarian Medical Research, Keio University School of Medicine. Written informed consent was obtained either from the participant or from a proxy if the participant lacked the capacity to provide consent. The ethics committee approved all cohort studies of the Keio University School of Medicine (ID: 20021020, 20022020, 20070047, and 20160297). The TOOTH and KAWP studies are registered in the University Hospital Medical Information Network Clinical Trial Registry (ID: UMIN000001842 and UMIN000026053).

### Baseline examination

All participants were examined by experienced geriatricians at the time of enrollment, following previously described protocols (*Arai et al., 2015*; *Arai et al., 2014*; *Gondo et al., 2006*; *Takayama et al., 2007*; *Arai et al., 2010*). Our assessment considered medical histories, lifestyle factors, and physical and cognitive functions. A mini-mental state examination (MMSE; 0–30 points) was used to assess cognitive function. The five-item World Health Organization well-being index (WHO5; 0–5 points) was used to assess current mental well-being. Instrumental activities of daily living (IADLs) were assessed using the Lawton scale (0–5 points) and independent IADL was defined as 5 points on the Lawton scale. The concentration of blood biomarkers, including cHMW adiponectin, N-terminal pro-brain natriuretic peptide (NTproBNP), cystatin C, and interleukin-6 (IL-6), was measured according to previously described protocols (*Hirata et al., 2020*). Blood test results for high-density lipoprotein cholesterol (HDLC), low-density lipoprotein cholesterol (LDLC), total cholesterol (TCHO), triglyceride (TG), choline esterase (CHE), aspartate aminotransferase (AST), γ-glutamyl transpeptidase (γGTP), lactate dehydrogenase (LDH), uric acid (UA), albumin (ALB), and HbA1c content were also obtained using previously described protocols (*Hirata et al., 2020*). A person with diabetes mellitus (DM) was defined as follows: individuals with glycated hemoglobin (HbA1c)≥6.5%, those receiving antidiabetic drug therapy, or those receiving insulin injections (*Supplementary file 1*).

### Measurement of cHMW adiponectin levels

The plasma cHMW adiponectin levels were measured using the Human HMW Adiponectin/Acrp30 Immunoassay Quantikine ELISA Kit (R&D Systems, Inc, Minneapolis, MN, USA) according to the manufacturer's protocol.

### Whole-genome DNA sequencing

Total genomic DNA was extracted from whole blood using a FlexGene DNA Kit (QIAGEN, Hilden, Germany). The whole-genome DNA sequence of 440 centenarians was determined using whole-genome DNA sequencing with previously described protocols (*Sasaki et al., 2021a*).

## Genotyping using DNA microarray and imputation

The genotypes of 0.65 M single nucleotide variants (SNVs) of 367 centenarians were determined using an Axiom Japonica Array NEO according to the manufacturer's protocol. The genotypes of 0.65 M SNVs of 1015 individuals in the KAWP study were determined using an Infinium Asian Screening Array-24 v1.0 BeadChip kit according to the manufacturer's protocol. All DNA microarray scan images were analyzed using previously described protocols (*Sasaki et al., 2021a*).

## Meta-quantitative trait association analysis for cHMW adiponectin level

To identify cHMW adiponectin level-associated SNVs in the very old and centenarians, we analyzed the association among cHMW adiponectin level and genetic variants using quantitative trait association analysis with the PLINK program (version 1.90) adjusted for sex and age at entry against 440 WGS and DNA microarray-imputed data for 367 centenarians and 1015 very old, respectively (*Purcell et al., 2007*). These quantitative trait association analyses were meta-analyzed using Metal (released on May, 5, 2020) (*Willer et al., 2010*). Finally, we obtained meta-quantitative trait association data between 5.75 M SNVs and cHMW adiponectin levels for 1822 individuals. A Manhattan plot was created using the qqman package (version 0.1.8) in program R (*Turner, 2018*). An enlarged view of a Manhattan plot with recombination rate information was generated using LocusZoom (version 1.3) (*Pruim et al., 2010*).

## Genotyping and minor allele frequency of rs4783244 (*CDH13*) and rs11711353 (*ADIPOQ*)

To determine the rs4783244 and rs11711353 genotypes and minor allele frequency in the very old and centenarians, we genotyped these SNVs using the TaqMan SNP Genotyping Assay system according to the manufacturer's protocols.

Minor allele frequency of rs4783244 and rs11711353 for Japanese controls (ToMMo 38KJPN) was used in the jMorp database (https://jmorp.megabank.tohoku.ac.jp).

## LASSO and multivariate analysis

For Least Absolute Shrinkage and Selection Operator (LASSO) and further multivariate analysis, cHMW adiponectin level was used as the outcome, and LASSO was used to evaluate 32 factors by LASSO including age at entry, BMI, systolic blood pressure (SBP), years of education, smoking history, IADL score, hand grip, cognitive impairment (MMSE: ≤23), WHO5 score, self-reported disease histories (heart disease, diabetes, cancers, renal disease, fracture), biomarkers in blood (HDLC, TCHO, LDLC, TG, CHE, AST, ALT, γGTP, LDH, UA, ALB, CstC, NTproBNP, HbA1c, IL-6), and genetic factors (sex, CDH13 rs4783244, ADIPOQ rs11711353) for the very old, and 26 factors including age at entry, BMI, SBP, five educational category, smoking history, activities of daily living (ADLs) score, self-reported disease histories (heart disease, diabetes, cancers, renal disease, fracture), biomarkers in blood (HDLC, TCHO, LDLC, TG, CHE, γGTP, UA, ALB, CstC, NTproBNP, HbA1c, IL-6), and genetic factors (sex, *CDH13* rs4783244, *ADIPOQ* rs11711353) for centenarians (*Supplementary file 1*). After excluding samples with any missing values in the selected factors, 1314 very old and 352 centenarians were selected.

## Survival analysis

For survival analysis, all-cause, cancer-case, CVD-cause, and pneumonia-cause mortalities were used as outcome, and BMI, cHMW adiponectin level, disease history (DM), number of allele for *CDH13* rs4783244 and *ADIPOQ* rs11711353, age at entry, HDLC, and years of education were used as potential confounder and effect modifiers based on the results of multivariate analysis. The very old and centenarians were grouped into three quantile cHMW adiponectin level groups (high, intermediate, and low) against 1425 very old (678 men and 747 women) and 545 centenarians (90 men and 455 women) for whom survival time information was available.

## Statistical analyses

Baseline characteristics, medical history, plasma biomarkers, and genotype data are expressed as a median or number with a percentage or IQR. The difference in baseline data was evaluated using Wilcoxon rank-sum, chi-square, and Fisher's exact tests (*Supplementary file 1*). Multivariate logistic

regression analyses were performed using a generalized linear model with factors selected by LASSO. All statistical analyses were performed using program R (version 4.0.3) with exactRankTests (wilcox. exact, Wilcoxon rank-sum test [version 0.8-31]), glmnet (LASSO and multivariate analyses [version 4.1]), survival (survival analysis (survfit, coxph, and cox.zph) [version 3.2-13]), powerSurvEpi (statistic power calculation [version 0.1.3]), and default packages.

## Results

### Baseline characteristics of the very old and centenarian cohorts

This study used data collected from prospective cohort studies, including the TOOTH and KAWP for the very old (aged 85–89 years) as well as the TCS and JSS for centenarians (aged 100 years and older, *Figure 1a*; *Arai et al., 2015*; *Arai et al., 2014*; *Gondo et al., 2006*; *Takayama et al., 2007*; *Arai et al., 2010*; *Arai et al., 2021*; *Sasaki et al., 2021b*). The data for cHMW adiponectin levels were available for 812 centenarians (woman: 84.4%, 87.7% in Japanese census data in 2020) and 1498 very old (woman: 51.7%, 64.4% in Japanese census data in 2020, *Figure 1—figure supplement 1*). Participant characteristics at enrollment are presented in *Supplementary file 1*. The flowchart for the analysis is shown in *Figure 1—figure supplement 2*.

cHMW adiponectin levels increased with age from 30 to 70 years and are higher in women than those in men (*Cnop et al., 2003*). Our findings in this study were consistent in that cHMW adiponectin levels gradually increased with age (*Figure 1b*; also observed in the longitudinal data of the TOOTH study, *Figure 1—figure supplement 3*), with a similar difference observed between sexes of the very old and centenarians (*Figure 1c and d*).

### Single nucleotide variations in the promoter regions of *CDH13* and *ADIPOQ* were associated with cHMW adiponectin levels in the very old and centenarians

A previous genome-wide association study (GWAS) has revealed that cHMW adiponectin levels are associated with two major loci, including *CDH13* (also called T-cadherin) and *ADIPOQ* (gene corresponding to adiponectin), both in European and multi-ethnic cohorts (*Dastani et al., 2012*). To confirm this association in the very old and centenarians, we quantitatively assessed 5.75 M SNVs adjusted for age at entry and sex from the genome data for 1822 individuals, including whole-genome DNA sequences for 440 centenarians, imputed microarray analysis data for 367 centenarians, and imputed microarray analysis data for 1015 very old (*Figure 2a*, *Figure 2—figure supplement 1*). We found that rs12051213 T>C SNV, located near exon 1 of *CDH13*, was the locus most significantly associated with cHMW adiponectin levels ($p=2.21 \times 10^{-22}$, Z score = –9.73), and rs11711353 A>G SNV, located near exon 1 of *ADIPOQ*, was the second-most significant locus ($p=5.72 \times 10^{-7}$, Z score = 5.00). The GWAS results also revealed that rs12051213, rs11711353, and other associated SNVs were mainly located around exon 1, indicating that these variants would be associated with the expression of *CDH13* and *ADIPOQ* genes (*Figure 2b and c*). For the *CDH13* locus, rs4783244 ($p=5.39 \times 10^{-22}$, Z score = –9.64) was located near rs12051213, another SNV commonly used as cHMW adiponectin level-associated SNV; therefore, we selected rs4783244 as a representative SNV among *CDH13*-associated SNVs. To confirm the association between these SNVs and cHMW adiponectin levels, we determined the genotype of these two SNVs against the very old and centenarians using a TaqMan assay. As a result, no significant difference in minor allele frequency was found between Japanese control (ToMMo 38KJPN), the very old, and centenarian men and women using Fisher's exact test and multiple testing (*Figure 2—figure supplement 2*). We compared the genotype-based distribution of cHMW adiponectin levels by genotype (*Figure 2d*, *Figure 2—figure supplement 2*). cHMW adiponectin levels were found to vary significantly between the rs4783244 reference allele homozygote and rs4783244 alternative allele heterozygote both in the very old and centenarians. However, except for very old men, no significant difference was observed between the rs4783244 alternative allele heterozygote and rs4783244 alternative allele homozygote in the very old or centenarians. Additionally, cHMW adiponectin levels varied significantly among several allele combinations of rs11711353 in very old or centenarian women but not in very old or centenarian men (*Figure 2—figure supplement 3*). These data indicated that both major loci (rs4783244 of *CDH13* and rs11711353 of *ADIPOQ*)

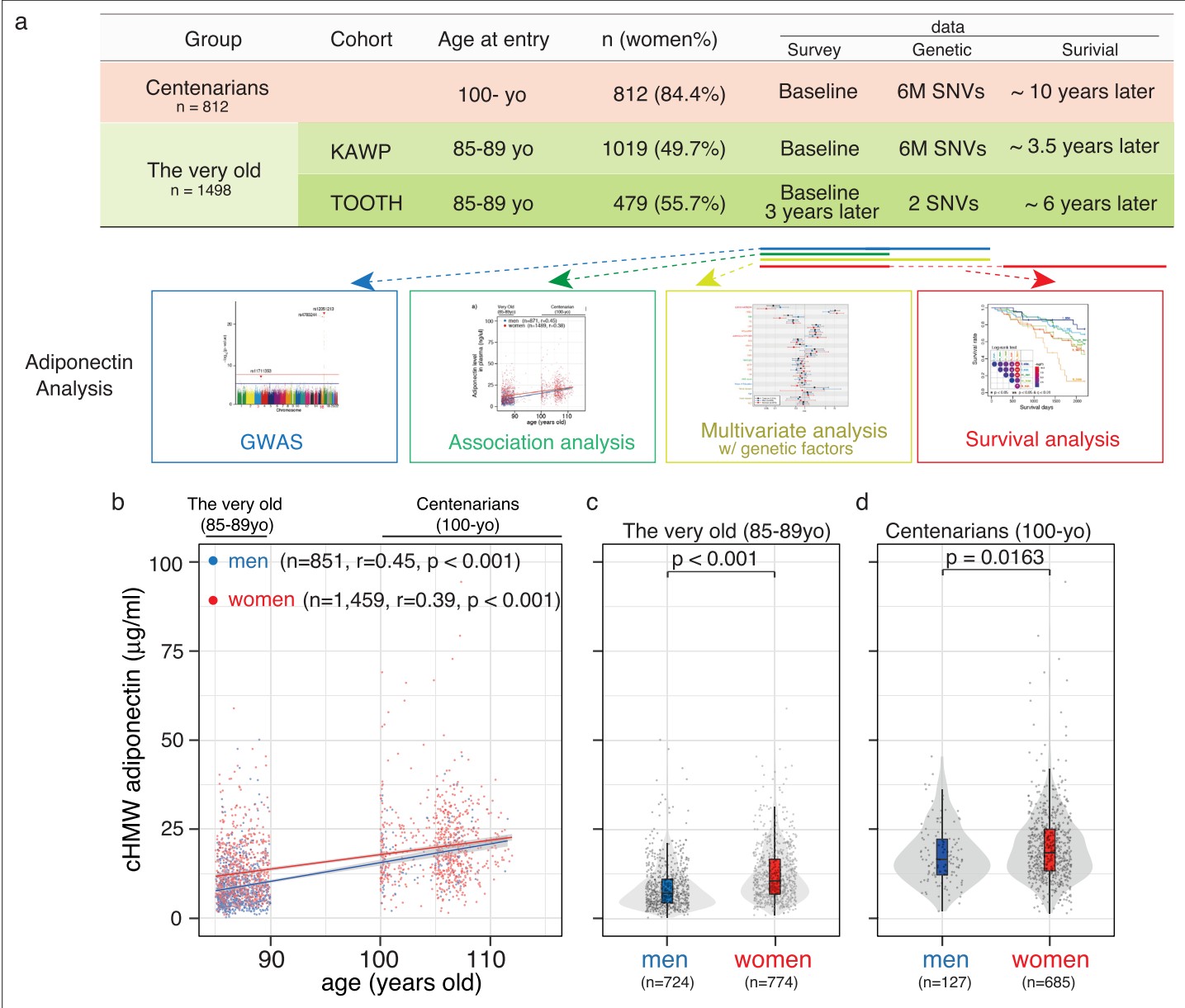

**Figure 1.** Analysis workflow and distribution of circulating high-molecular-weight (cHMW) adiponectin levels in the very old and centenarians. (a) Sample summary and analysis workflow of cHMW adiponectin levels. (b) Distribution of cHMW adiponectin levels in older adults and centenarians. cHMW adiponectin levels gradually increased with age in the very old to centenarians. (c) Distribution of cHMW adiponectin levels in older men and women. (d) Distribution of cHMW adiponectin levels in centenarian men and women. The difference in cHMW adiponectin levels was significant between sexes in both the very old and centenarians.

The online version of this article includes the following source data and figure supplement(s) for figure 1:

**Source data 1.** Source data for *Figure 1* including 812 centenarians and 1498 very old data.

**Figure supplement 1.** Description of the cohorts in this study.

**Figure supplement 2.** Flowchart for analysis in this study.

**Figure supplement 3.** Transition of circulating high-molecular-weight (cHMW) adiponectin level in the longitudinal data of Tokyo Oldest Old Survey on Total Health (TOOTH) study.

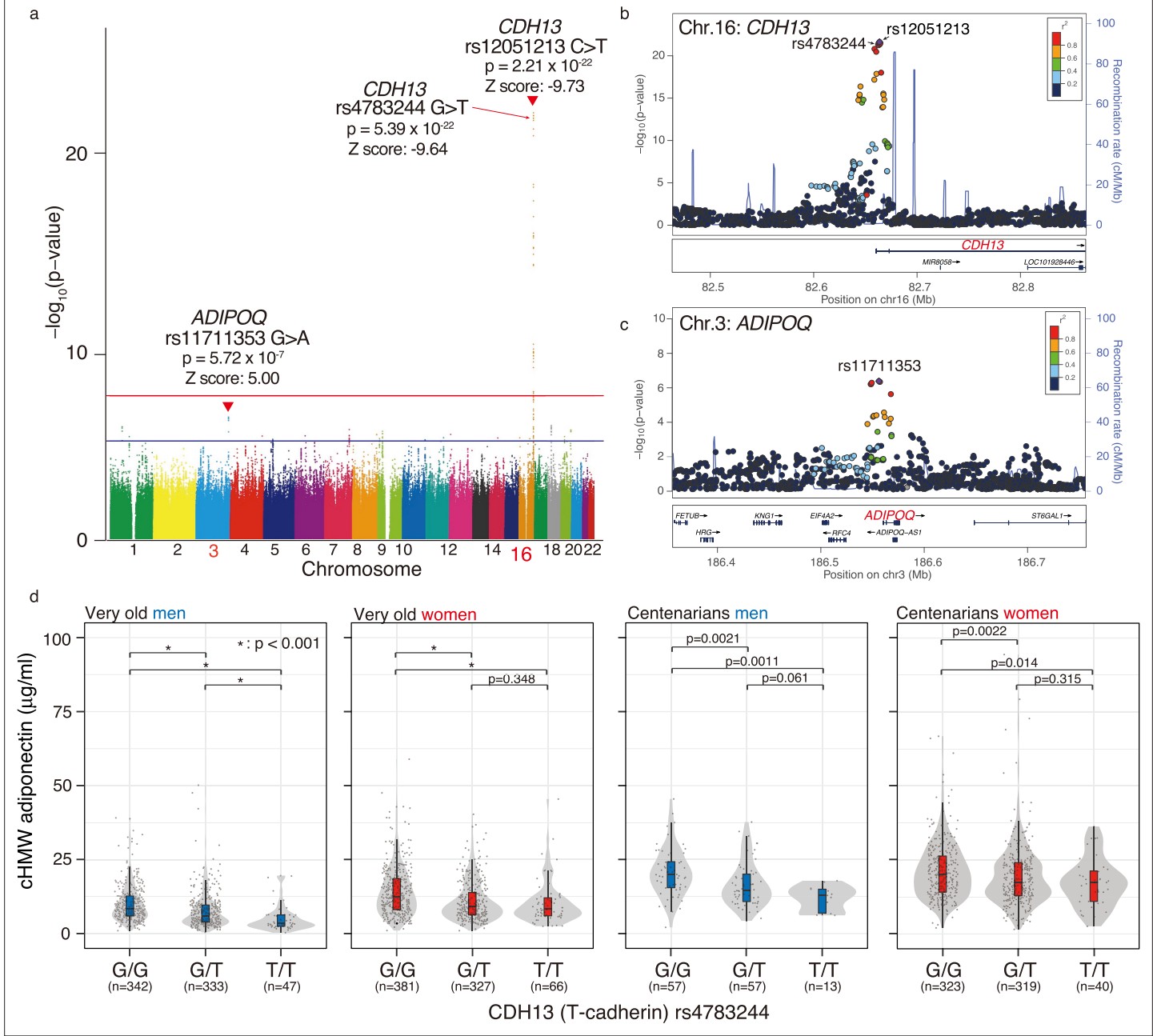

**Figure 2.** Meta-genome-wide association study (GWAS) for circulating high-molecular-weight (cHMW) adiponectin levels in the very old and centenarians. (**a**) Meta-GWAS analysis for cHMW adiponectin levels in the very old and centenarians. Number of samples for the very old and centenarian were 1015 and 807, respectively. Loci for *CDH13* (rs12051213, C: reference allele, T: alternative allele, $p=2.45 \times 10^{-22}$) and *ADIPOQ* (rs11711353, G: reference allele, A: alternative allele, $p=6.68 \times 10^{-7}$) were detected using meta-GWAS for cHMW adiponectin levels in older adults and centenarians. (**b**) A GWAS enlarged view of the *CDH13* region. (**c**) A GWAS enlarged view of the *ADIPOQ* region. (**d**) Distribution of cHMW adiponectin levels in rs4783244 (*CDH13*) genotypes of the very old and centenarians. cHMW adiponectin levels varied significantly between the rs4783244 reference allele homozygote and rs4783244 alternative allele heterozygote in the very old and centenarians. Except in very old men, no significant difference was observed between the rs4783244 alternative allele heterozygote and rs4783244 alternative allele homozygote in the very old or centenarians.

The online version of this article includes the following source data and figure supplement(s) for figure 2:

**Source data 1.** Source data for *Figure 1* including 812 centenarians and 1498 very old data.

**Figure supplement 1.** Genome-wide association study (GWAS) for circulating high-molecular-weight (cHMW) adiponectin level.

**Figure supplement 2.** Minor allele frequency comparison of rs4783244 (CDH13) and rs11711353 (ADIPOQ).

**Figure supplement 3.** Distribution of circulating high-molecular-weight (cHMW) adiponectin level in rs11711353 (ADIPOQ) genotypes of the very old and centenarians.

were associated with cHMW adiponectin levels in the very old and centenarians. However, the effects depended on age and sex.

## The characteristics of cHMW adiponectin levels in the very old and centenarians

A previous study reported a negative association between cHMW adiponectin levels and T2DM prevalence and BMI, as well as a positive association with insulin sensitivity index, TG content, and HDLC levels (*Li et al., 2009*; *Cnop et al., 2003*). To evaluate these associations in the oldest old, we analyzed the association between cHMW adiponectin levels and DM, HDLC, and BMI (*Figure 3*). A person with DM was defined as follows: individuals with glycated hemoglobin (HbA1c)≥6.5%, those receiving anti-diabetic drug therapy, or those receiving insulin injections (*Figure 3a and b*). We found that cHMW adiponectin levels in the DM group were significantly lower than those in the non-DM group in both the very old and centenarians, indicating that adiponectin is associated with the DM pathway, regardless of age.

Although blood-lipid contents, including TCHO, HDLC, LDLC, and TG gradually decreased with age from the very old to centenarians (*Supplementary file 1*), a positive association was observed between cHMW adiponectin and HDLC levels (*Figure 3c, d*). A negative association between cHMW adiponectin levels and BMI was observed in the very old, though this association was less prominent in centenarians (*Figure 3e, f*). These findings suggested that the physiological factors associated with adiponectin may vary from the very old to centenarians.

## The factors associated with cHMW adiponectin levels vary between the very old and centenarians

In our multivariate regression analysis of cHMW adiponectin levels, we initially selected 32 factors for the very old, including cHMW adiponectin level-associated genetic factors (genotypes of rs4783244 in *CDH13* and rs11711353 in *ADIPOQ*), and 26 factors for centenarians based on a previous report (*Cnop et al., 2003*; *Muratsu et al., 2021*). To reduce the effects of multicollinearity, we used a LASSO method with fivefold cross-validation and identified 19 factors for the very old and 7 factors for centenarians (*Figure 4—figure supplement 1*). According to the multivariate regression analysis for the very old, 14 significant factors for men and 10 significant factors for women were identified (*Figure 4a*, *Supplementary files 2–4*); among centenarians, three significant factors for men and four significant factors for women were identified (*Figure 4b*, *Supplementary files 5–7*). An analysis of deviance revealed that the total variance of known cHMW adiponectin level-associated factors was 36.8–42.0% in the very old and centenarian men and 18.4% in centenarian women (*Figure 4c, d*, *Supplementary files 8–11*). These results suggest that the genotypes of rs4783244 in *CDH13*, HDLC, BMI, and lipid metabolism-associated factors, including HDLC and TG, are major factors associated with cHMW adiponectin levels in both sexes of the very old. Furthermore, the genotypes of rs4783244 in *CDH13* and HDLC were also associated with cHMW adiponectin levels in centenarians. Significantly, the current known factors associated with cHMW adiponectin levels were expected to correspond to 18.4% of the total variance in centenarian women, indicating a reduced contribution of known factors associated with cHMW adiponectin levels in centenarians. Thus, major cHMW adiponectin-associated factors found in the very old would not be responsible for the age-dependent increment of cHMW adiponectin levels.

## Higher cHMW adiponectin levels in very old men was positively associated with high all-cause mortality rates, independent of BMI

High cHMW adiponectin levels are associated with increased all-cause mortality and CVD risk in adults in their 60s and 70s (*Kizer et al., 2008*; *Poehls et al., 2009*; *Kim-Mitsuyama et al., 2019*; *Choi et al., 2015*). To evaluate the effects of cHMW adiponectin levels on mortality in the very old and centenarians, hazard ratios of all-cause mortality were analyzed using Cox promotional hazards models for three quantiles of cHMW adiponectin levels (i.e., high, intermediate, and low) in 1425 very old (678 men and 747 women) and 545 centenarians (90 men and 455 women) for whom both survival time information and a number of covariates were available. Prior to the analysis, the availability of sufficient samples and events for all-cause mortality were ensured for the survival analysis of the very old and centenarian women and there was no significant difference in the proportional hazards assumption of the cHMW

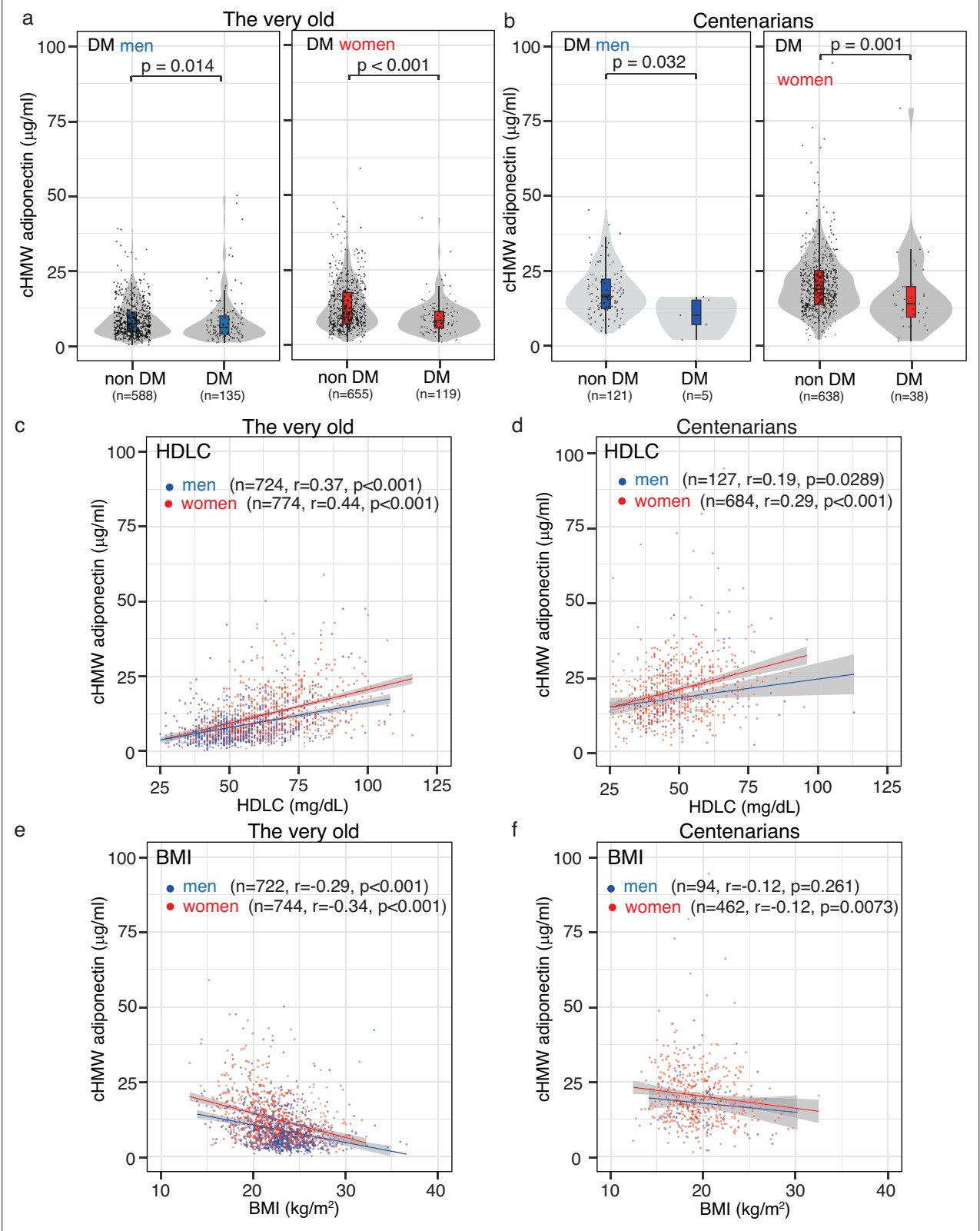

**Figure 3.** Association between circulating high-molecular-weight (cHMW) adiponectin level, high-denisty lipoprotein cholesterol (HDLC), body mass index (BMI), and glycated hemoglobin (HbA1c). (**a, b**) Distribution of cHMW adiponectin levels in the diabetes mellitus (DM) and non-DM groups. A person with DM was defined as follows: individuals with glycated hemoglobin (HbA1c)≥6.5%, those receiving antidiabetic drug therapy, or those receiving insulin injections. cHMW adiponectin levels in the DM group were significantly lower than those in the non-DM group in the very old

*Figure 3 continued on next page*

*Figure 3 continued*

and centenarians. (**c, d**) Association between cHMW adiponectin levels and HDLC content. A positive association was observed between cHMW adiponectin levels and HDLC content in the very old and centenarians. (**e, f**) Association between cHMW adiponectin levels and BMI. A strong negative association was observed between cHMW adiponectin levels and BMI in the very old, though this association was rarely observed in centenarians.

The online version of this article includes the following source data for figure 3:

**Source data 1.** Source data for *Figure 1* including 812 centenarians and 1498 very old data.

adiponectin level and each of the covariates (*Figure 5—figure supplements 1 and 2*). However, the statistical power analysis indicated that there were not sufficient events, and samples were ensured for the centenarian men even if they were divided into two groups. Within the follow-up periods, 145 (21.3%) men and 101 (13.5%) women died in the very old, whereas 89 (98.9%) men and 542 (99.4%) women died in the centenarians (*Figure 5* and *Supplementary file 1*). As a result, the hazard ratios of all-cause mortality for intermediate and high levels of cHMW adiponectin groups in very old men were significantly higher (HR: 1.67 and 2.32) rather than those of the all-cause mortality of the low cHMW adiponectin level group (reference), even after adjustment for BMI (HR: 1.60 and 2.12). In contrast, the hazard ratio for all-cause mortality for the high cHMW adiponectin levels group in very old women was

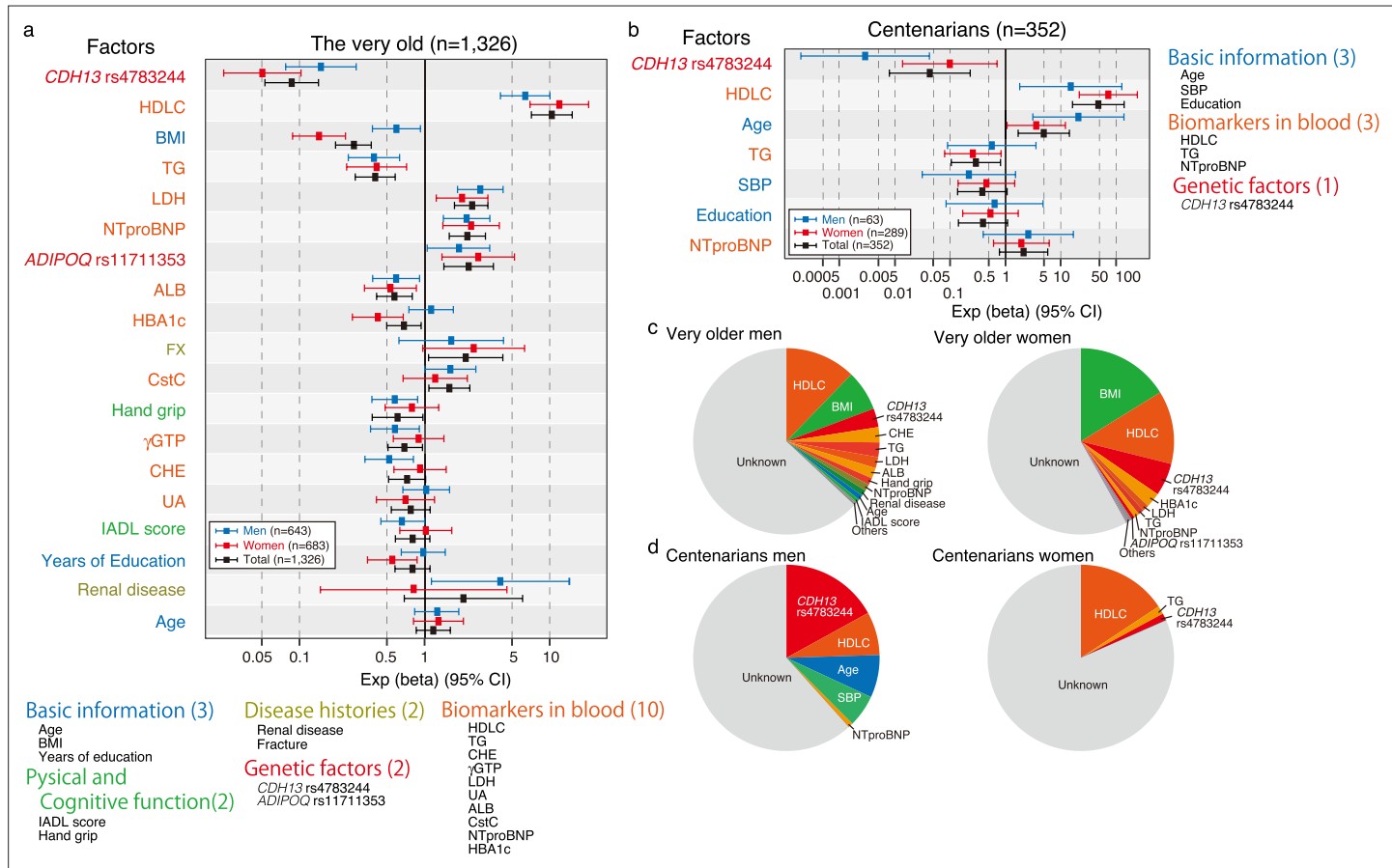

**Figure 4.** Multivariate analysis for circulating high-molecular-weight (cHMW) adiponectin levels in the very old and centenarians. (**a**) Multivariate analysis for cHMW adiponectin levels in very old men and women; 14 significant factors for very old men and 10 significant factors for very old women were identified. (**b**) Multivariate analysis for cHMW adiponectin levels in centenarian men and women; 3 significant factors for centenarian men and 4 significant factors for centenarian women were identified. (**c**) The contribution rate for each factor in very old men and women was estimated by analysis of variance. (**d**) The contribution rate for each factor in centenarian men and women was estimated using analysis of variance. The total variance of known cHMW adiponectin level associated factors corresponded to 36.8–42.0% in very old and centenarian men and 18.4% in centenarian women.

The online version of this article includes the following figure supplement(s) for figure 4:

**Figure supplement 1.** Least Absolute Shrinkage and Selection Operator (LASSO) with fivefold cross-validation against 1326 very old and 352 centenarians.

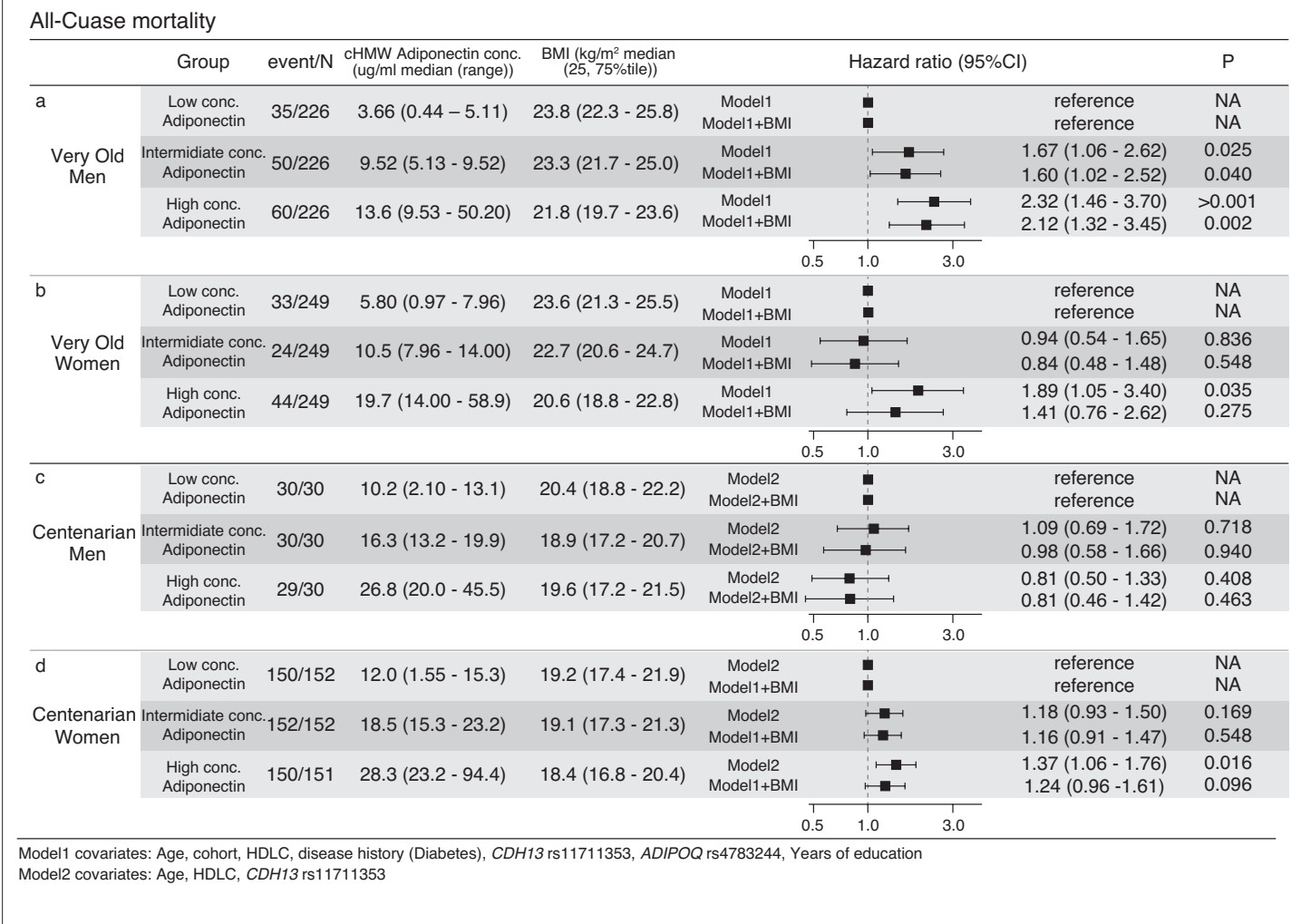

All-Cuase mortality

| | Group | event/N | cHMW Adiponectin conc. (ug/ml median (range)) | BMI (kg/m² median (25, 75%tile)) | | Hazard ratio (95%CI) | P |
|---|---|---|---|---|---|---|---|
| **a** Very Old Men | Low conc. Adiponectin | 35/226 | 3.66 (0.44 – 5.11) | 23.8 (22.3 - 25.8) | Model1 / Model1+BMI | reference / reference | NA / NA |
| | Intermidiate conc. Adiponectin | 50/226 | 9.52 (5.13 - 9.52) | 23.3 (21.7 - 25.0) | Model1 / Model1+BMI | 1.67 (1.06 - 2.62) / 1.60 (1.02 - 2.52) | 0.025 / 0.040 |
| | High conc. Adiponectin | 60/226 | 13.6 (9.53 - 50.20) | 21.8 (19.7 - 23.6) | Model1 / Model1+BMI | 2.32 (1.46 - 3.70) / 2.12 (1.32 - 3.45) | >0.001 / 0.002 |
| **b** Very Old Women | Low conc. Adiponectin | 33/249 | 5.80 (0.97 - 7.96) | 23.6 (21.3 - 25.5) | Model1 / Model1+BMI | reference / reference | NA / NA |
| | Intermidiate conc. Adiponectin | 24/249 | 10.5 (7.96 - 14.00) | 22.7 (20.6 - 24.7) | Model1 / Model1+BMI | 0.94 (0.54 - 1.65) / 0.84 (0.48 - 1.48) | 0.836 / 0.548 |
| | High conc. Adiponectin | 44/249 | 19.7 (14.00 - 58.9) | 20.6 (18.8 - 22.8) | Model1 / Model1+BMI | 1.89 (1.05 - 3.40) / 1.41 (0.76 - 2.62) | 0.035 / 0.275 |
| **c** Centenarian Men | Low conc. Adiponectin | 30/30 | 10.2 (2.10 - 13.1) | 20.4 (18.8 - 22.2) | Model2 / Model2+BMI | reference / reference | NA / NA |
| | Intermidiate conc. Adiponectin | 30/30 | 16.3 (13.2 - 19.9) | 18.9 (17.2 - 20.7) | Model2 / Model2+BMI | 1.09 (0.69 - 1.72) / 0.98 (0.58 - 1.66) | 0.718 / 0.940 |
| | High conc. Adiponectin | 29/30 | 26.8 (20.0 - 45.5) | 19.6 (17.2 - 21.5) | Model2 / Model2+BMI | 0.81 (0.50 - 1.33) / 0.81 (0.46 - 1.42) | 0.408 / 0.463 |
| **d** Centenarian Women | Low conc. Adiponectin | 150/152 | 12.0 (1.55 - 15.3) | 19.2 (17.4 - 21.9) | Model2 / Model1+BMI | reference / reference | NA / NA |
| | Intermidiate conc. Adiponectin | 152/152 | 18.5 (15.3 - 23.2) | 19.1 (17.3 - 21.3) | Model2 / Model1+BMI | 1.18 (0.93 - 1.50) / 1.16 (0.91 - 1.47) | 0.169 / 0.548 |
| | High conc. Adiponectin | 150/151 | 28.3 (23.2 - 94.4) | 18.4 (16.8 - 20.4) | Model2 / Model1+BMI | 1.37 (1.06 - 1.76) / 1.24 (0.96 -1.61) | 0.016 / 0.096 |

Model1 covariates: Age, cohort, HDLC, disease history (Diabetes), *CDH13* rs11711353, *ADIPOQ* rs4783244, Years of education
Model2 covariates: Age, HDLC, *CDH13* rs11711353

**Figure 5.** Survival analysis using Cox promotional hazards model for three quantile circulating high-molecular-weight (cHMW) adiponectin level groups. (**a, b**) Survival analysis of very old men and women using the Cox promotional hazards model for three quantile cHMW adiponectin level groups. Seven covariates (model1) and seven covariates with body mass index (BMI) were used for calculating the multiple regression analysis of Cox promotional hazards model. Hazard ratio for low concentration adiponectin group was calculated as the reference. The statistics power analysis using powerSurvEpi (version 0.1.3) indicated that survival analyses for both very old men and women have sufficient number of samples and events. (**c, d**) Survival analysis of the centenarian men and women using Cox promotional hazards model for three quantile cHMW adiponectin level groups. Three covariates (model2) and three covariates with BMI were used for calculation of multiple regression analysis of Cox promotional hazards model. Hazard ratio for low concentration adiponectin group was calculated as the reference. The statistics power analysis using powerSurvEpi (version 0.1.3) indicated that survival analysis for centenarian women has sufficient number of samples and events, however, survival analysis for centenarian men was underpowered due to insufficient number of events.

The online version of this article includes the following figure supplement(s) for figure 5:

**Figure supplement 1.** The proportional hazards assumption test for a Cox regression model fit.

**Figure supplement 2.** The proportional hazards assumption test for a Cox regression model fit.

**Figure supplement 3.** Survival time analysis for cancer-cause mortality against the three quantile groups of circulating high-molecular-weight (cHMW) adiponectin levels in very old men and women.

**Figure supplement 4.** Survival time analysis for cardiovascular disease-cause mortality against the three quantile groups of circulating high-molecular-weight (cHMW) adiponectin levels in very old men and women.

**Figure supplement 5.** Survival time analysis for pneumonia-cause mortality against the three quantile groups of circulating high-molecular-weight (cHMW) adiponectin levels in very old men and women.

**Figure supplement 6.** J-CHS frailty index distribution against the three quantile groups of circulating high-molecular-weight (cHMW) adiponectin levels and multiple regression analysis in very old men and women (Kawasaki Aging Wellbeing Project [KAWP]).

*Figure 5 continued*

**Figure supplement 7.** Adiponectin mRNA expression analysis of single-cell RNA-seq for four kinds of mouse adipose tissue.

**Figure supplement 8.** Ectopic expression of adiponectin with aging in mouse.

significantly higher (HR: 1.89), but was not significant after adjustment for BMI (HR: 1.41, *Figure 5*). This trend was also observed in the centenarian women.

To further elucidate the factors associated with mortality, we also analyzed cause-specific mortality associated with cancer, CVD, and pneumonia in the very old (*Figure 5—figure supplements 3–5*). The total number of events for each cause-specific mortality was 59 (cancer), 53 (CVD), and 40 (pneumonia), indicating that the analysis lacked sufficient statistical power. Testing populations with a 5% difference in event frequency would require approximately 440 samples for each group.

## Discussion

The results of this study showed that cHMW adiponectin levels increased with age up to centenarians, although the associated factors varied with sex. Therefore, we are further elucidating whether the increment of cHMW adiponectin level with age extends into very old and exceptionally old age. Meta-GWAS with cHMW adiponectin levels revealed that the SNVs of two loci containing the promoter regions of *CDH13* and *ADIPOQ* genes were associated with cHMW adiponectin levels. The levels of HDLC were associated with those of cHMW adiponectin both in the very old and centenarians, though the association with BMI was relatively weaker in centenarians. The multivariate regression analysis with factor selection using the LASSO method revealed that genetic variants, BMI, and lipids were major factors associated with cHMW adiponectin level in the very old; here, BMI was not selected as an associated factor in centenarians. The analysis of deviance revealed that the contribution of known factors to cHMW adiponectin levels decreased in centenarian women, suggesting that the major factors in the very old would not be responsible for the age-dependent increase in cHMW adiponectin levels. The high cHMW adiponectin levels in very old men were associated with all-cause mortality independently of BMI; however, no association was observed between the cHMW adiponectin levels and all-cause mortality in very old and centenarian women. Therefore, the contribution of known major factors associated with cHMW adiponectin levels, including BMI and lipid content, varies with age, suggesting that its physiological significance also varies with age in the oldest old.

The salutary effects of adiponectin on glucose homeostasis, insulin sensitivity, and chronic low-grade inflammation, and the inverse association between the incidence of T2DM and cHMW adiponectin levels are known (*Kadowaki et al., 2006*; *Lindsay et al., 2002*; *Spranger et al., 2003*). We have previously reported that a low incidence of T2DM is a characteristic of centenarians; therefore, we deduced that the high cHMW adiponectin levels in centenarians would be partially influenced by a low incidence of T2DM. In the present study, the T2DM group showed significantly lower levels of cHMW adiponectin, regardless of the cohort, suggesting the physiological significance of cHMW adiponectin levels in the context of insulin sensitivity and T2DM incidence is consistent across ages.

We revealed that very old men with high cHMW adiponectin levels show high rates of all-cause mortality, consistent with previous reports for adults in their 60s and 70s (*Kizer et al., 2008*; *Poehls et al., 2009*; *Kim-Mitsuyama et al., 2019*; *Choi et al., 2015*). Moreover, cHMW adiponectin levels were associated with all-cause mortality independently of BMI. Excess weight loss can cause frailty in the oldest old, exacerbating mortality rates and death due to pneumonia (*Falcone et al., 2012*). Based on these results, we deduced that a combination of high cHMW adiponectin levels and low BMI may exert synergistic effects in the mortality among very old men. We also revealed that high cHMW adiponectin levels were not associated with mortality both in very old and centenarian women. Surprisingly, strength of the association between BMI and cHMW adiponectin level decreased in centenarians. Although the major factors associated with cHMW adiponectin level in centenarians were unknown, these results suggest that the factors associated with cHMW adiponectin levels vary with age, which would also alter the physiological significance of cHMW adiponectin level as it relates to mortality.

Frailty is an important concept in health maintenance and the process of functional decline in the oldest old. Recently, plasma adiponectin levels have been positively associated with frailty in the oldest old (*Nagasawa et al., 2018*; *Lee et al., 2021*). In our cohort, most centenarians were classified as frail

according to the current frailty criteria, so it is difficult to assess frailty in centenarians. For the very old, only the KAWP, one of the cohorts that included the very old, collected sufficient data to assess frailty. Using these limited data for the very old, we analyzed the distribution of cHMW adiponectin levels in each frailty category and analyzed their association with the revised J-CHS frailty index criteria using multiple regression analysis (*Satake and Arai, 2020*). As a result, we found that cHMW adiponectin levels were significantly associated with frailty, both in very old men and women (*Figure 5—figure supplement 6*). The cHMW adiponectin level was also significantly associated with frailty in very old women even after adjustment for BMI; however, no significant association was observed in very old men after adjustment by BMI. Thus, cHMW adiponectin levels would be associated with frailty in the very old, especially in women.

Although cHMW adiponectin levels increased with age, their association with BMI was comparatively lower in centenarians than that in the very old. This raises the question of which cells are responsible for the increased expression of adiponectin with aging. One hypothesis is that the clearance mechanism of adiponectin from the blood may be impaired by reduced kidney function, resulting in an accumulation of cHMW adiponectin. However, we did not observe a significant association between the levels of cHMW adiponectin and plasma cystatin C, one of the kidney function markers. Another hypothesis is that aging would cause ectopic *ADIPOQ* gene expression, increasing cHMW adiponectin levels. Re-analysis of in silico mouse single-cell transcriptomic data revealed that a small number of cells derived from subcutaneous adipose tissue expressed high levels of *ADIPOQ,* including brown, gonadal, mesenteric, and subcutaneous adipose tissues (*Figure 5—figure supplement 7*; *Tabula Muris Consortium, 2020*). Furthermore, a re-analysis of mouse whole-body single-cell transcriptomic data from 24 tissues during 1–30 months of age revealed that *ADIPOQ* mRNA was rarely expressed in tissues other than the fat tissue, even at advanced ages of 24, 27, and 30 months (*Figure 5—figure supplement 8 Tabula Muris Consortium, 2020*). These findings indicate that no universal mechanism between humans and mice would exist to induce cHMW adiponectin through ectopic expression of the *ADIPOQ* gene by aging.

The study had the following limitations: (1) Surveys for centenarians tend to have many missing values due to their limited physical and cognitive function; therefore, multivariate analysis using a series of covariates tends to reduce the number of samples to be analyzed. (2) Although the short survival time of centenarian in this showed no association between cHMW adiponectin level and all-cause mortality in this study, strong factors associated with survival, such as NTproBNP and ALB, tend to be detectable, while weaker factors are more difficult to detect. (3) Cox regression for all-cause mortality in centenarian men and cause-specific mortality in the very old men was statistically underpowered due to the insufficient size of samples and/or events. CVD mortality in very old men showed a trend to be associated with cHMW adiponectin levels, but statistically, twice the number of events or twice the number of total samples are needed to assess this. (4) Analysis of cHMW adiponectin levels and frailty in centenarians is difficult because most centenarians would be classified as frail according to the current frailty criteria. Of the two cohort studies of very old participants, the TOOTH study did not have sufficient data adjusted for the evaluation of J-CHS frailty criteria. Therefore, the association between cHMW adiponectin levels and frailty was analyzed in selected samples derived only from the KAWP study. This was only a cross-sectional analysis, and further analysis would be needed to prove causality. Therefore, these are described only in the Discussion section.

In this study, we verified the association among cHMW adiponectin level, BMI, and all-cause mortality in the very old and centenarians. Due to changes in the physiological significance of BMI between young and old ages, the appropriate BMI value is expected to vary with age. While a low BMI is recommended at a young age due to the risk of diabetes and metabolic syndrome, a high (though not excessively high) BMI is recommended at a later stage of life to decrease the risk of frailty and mortality. Therefore, the biological significance of cHMW adiponectin levels would also be changed depending on the biological significance of BMI in the aging process. The reasons for the high cHMW adiponectin levels and loss of association with BMI in centenarians remain unknown; however, future research should focus on identifying cells that expressing adiponectin, which should clarify its physiological significance in the oldest old.

## Additional information

### Competing interests

Hideyuki Okano: received consulting fees from SanBio Co.Ltd and K Pharma Inc, and participates on the Advisory Board for both SanBio Co.Ltd and K Pharma Inc. The author is President of the Japanese Society for Regenerative Medicine and the Japanese Society for Neurochemistry. The author has no other competing interests to declare. Yasumichi Arai: has received a grant from the Cyclic Innovation for Clinical Empowerment (AMED EKID), and from DAIICHI SANKYO Co, Ltd. The author has no other competing interests to declare. The other authors declare that no competing interests exist.

### Funding

| Funder | Grant reference number | Author |
| --- | --- | --- |
| A GRANT-IN-AID FOR SCIENTIFIC RESEARCH | 21590775 | Nobuyoshi Hirose<br>Michiyo Takayama<br>Yasumichi Arai |
| A GRANT-IN-AID FOR SCIENTIFIC RESEARCH | 24590898 | Nobuyoshi Hirose<br>Yasumichi Arai |
| A GRANT-IN-AID FOR SCIENTIFIC RESEARCH | 15KT0009 | Yasumichi Arai |
| A GRANT-IN-AID FOR SCIENTIFIC RESEARCH | 18H03055 | Yasumichi Arai |
| A GRANT-IN-AID FOR SCIENTIFIC RESEARCH | 20K20409 | Takashi Sasaki<br>Yasumichi Arai |
| A GRANT-IN-AID FOR SCIENTIFIC RESEARCH | 20K07792 | Takashi Sasaki<br>Yasumichi Arai |
| A GRANT-IN-AID FOR SCIENTIFIC RESEARCH | 23H03337 | Takashi Sasaki<br>Yasumichi Arai |
| Japan Science and Technology Agency | JP15667051 | Toru Takebayashi<br>Yasumichi Arai |
| Japan Agency for Medical Research and Development | 16kk0205009h001 | Yasumichi Arai |
| Japan Agency for Medical Research and Development | 17jm0210051h0001 | Takashi Sasaki<br>Yasumichi Arai |
| Japan Agency for Medical Research and Development | 19dk0207045h0001 | Takashi Sasaki<br>Yasumichi Arai |
| Keio University Global Research Institute | | Hideyuki Okano<br>Yasumichi Arai |
| Kanagawa Institute of Industrial Science and Technology | | Yasumichi Arai |

The funders had no role in study design, data collection and interpretation, or the decision to submit the work for publication.

### Author contributions

Takashi Sasaki, Conceptualization, Data curation, Formal analysis, Funding acquisition, Validation, Investigation, Visualization, Writing – original draft; Yoshinori Nishimoto, Takumi Hirata, Data curation, Investigation, Writing – review and editing; Yukiko Abe, Data curation, Validation, Investigation; Nobuyoshi Hirose, Investigation; Michiyo Takayama, Investigation, Writing – review and editing; Toru Takebayashi, Supervision, Funding acquisition, Project administration, Writing – review and editing; Hideyuki Okano, Supervision, Project administration, Writing – review and editing; Yasumichi Arai, Conceptualization, Supervision, Investigation, Project administration, Writing – review and editing

## Author ORCIDs
Takashi Sasaki ![ORCID] https://orcid.org/0000-0002-6275-046X

## Ethics
Written informed consent was obtained either from the participant or from a proxy if the participant lacked the capacity to provide consent. The ethics committee approved all cohort studies of the Keio University School of Medicine (ID: 20021020, 20022020, 20070047, and 20160297). The TOOTH and KAWP studies are registered in the University Hospital Medical Information Network Clinical Trial Registry (ID: UMIN000001842 and UMIN000026053).

## Decision letter and Author response
Decision letter https://doi.org/10.7554/eLife.86309.sa1
Author response https://doi.org/10.7554/eLife.86309.sa2

---

## Additional files

### Supplementary files
• Supplementary file 1. Participants' characteristics at enrollment 1: Wilcoxon ranking test, 2: Fisher's exact test, 3: Chi-square test. Abbreviations: IQR, interquartile range; BMI, body mass index; SBP, systolic blood pressure; IADLs, instrumental activities of daily living; ADLs, activities of daily living; MMSE, mini-mental state examination; HMW, high molecular weight; HDLC, high-density lipoprotein cholesterol; LDLC, low-density lipoprotein cholesterol; TCHO, total cholesterol; TG, triglyceride; CHE, choline esterase; AST, aspartate aminotransferase; γGTP, γ-glutamyl transpeptidase; LDH, lactate dehydrogenase; UA, uric acid; ALB, albumin; Alt, alternative; MAF, minor allele frequency.

• Supplementary file 2. Coefficients for generalized linear model analysis of plasma HMW adiponectin level in very old men (n=643).

• Supplementary file 3. Coefficients for generalized linear model analysis of plasma HMW adiponectin level in very old women (n=683).

• Supplementary file 4. Coefficients for generalized linear model analysis of plasma HMW adiponectin level in the very old (n=1,326).

• Supplementary file 5. Coefficients for generalized linear model analysis of plasma HMW adiponectin level in centenarian men (n=63).

• Supplementary file 6. Coefficients for generalized linear model analysis of plasma HMW adiponectin level in centenarian women (n=289).

• Supplementary file 7. Coefficients for generalized linear model analysis of plasma HMW adiponectin level in centenarian (n=352).

• Supplementary file 8. Analysis of variance of plasma HMW adiponectin level by ANOVA in very old men (n=643).

• Supplementary file 9. Analysis of variance of plasma HMW adiponectin level by ANOVA in very old women (n=683).

• Supplementary file 10. Analysis of variance of plasma HMW adiponectin level by ANOVA in centenarian men (n=63).

• Supplementary file 11. Analysis of variance of plasma HMW adiponectin level by ANOVA in centenarian women (n=289).

• MDAR checklist

• Source code 1. R script code file for *Figure 1c*.

• Source code 2. R script code file for *Figure 1c*.

• Source code 3. R script code file for *Figure 2d*.

• Source code 4. R script code file for *Figure 3a, b*.

• Source code 5. R script code file for *Figure 3a, b*.

### Data availability
The cHMW adiponectin levels and covariates data were deposited with this manuscript as source data files. The data with age for the very old and centenarians have ethical and legal restrictions to public deposition due to avoid personal identification, and will be available upon request with an

appropriate research arrangement with approval of the Research Ethics Committee of Keio University School of Medicine for Clinical Research. To request, please contact Takashi Sasaki (corresponding author) via e-mail: sasasa@z5.keio.jp.

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
