## [Editor Report]

This study provides useful descriptive findings regarding levels of adiponectin and its associations with genetic variants and age in very old and centenarian men and women. Due to some missing data, the evidence supporting the conclusions about cross-sectional determinants of circulating levels of adiponectin in centenarians is incomplete. The evidence for conclusions about relationships between adiponectin and survival is also incomplete because of some missing data and unadjusted survival analyses.

---

## [Decision Letter]

**Decision letter after peer review:**

Thank you for submitting your article "Status and physiological significance of circulating adiponectin in the very old and centenarians: observation study" for consideration by *eLife*. Your article has been reviewed by 2 peer reviewers, and the evaluation has been overseen by a Reviewing Editor and Carlos Isales as the Senior Editor. The following individual involved in the review of your submission has agreed to reveal their identity: Sandra Aleksic (Reviewer #1).

The reviewers have discussed their reviews with one another, and the Reviewing Editor has drafted this to help you prepare a revised submission. We trust that the reviewers' feedback will be helpful to you in preparing the revision.

Essential revisions:

1) Conclusions about survival are not supported by data. Only one adjusted survival analysis is presented (very old men) and in that analysis adiponectin levels were not significantly associated with mortality after adjustment for BMI and other confounders. Other survival analyses are presented as unadjusted KM curves with log-rank tests, frequently with small numbers of participants and with crossing survival curves (questioning the assumption of proportional hazards).

2) One potentially interesting finding is a possible greater mortality in very old men with low BMI and high adiponectin, compared to other combinations of BMI and adiponectin tertiles, including low BMI and mid and low adiponectin, which suggests that it is possible that some of the association between adiponectin and mortality is independent of the BMI at that time point (literature suggests that mortality association of adiponectin is abolished when adjusted for weight loss and low muscle mass in older adults PMID: 29438496).

3) There is a substantial unexplained missingness of data. More than two-thirds of the participants in the very old group are missing mortality data and the reason for this was not provided. More than one-half of centenarians are not included in multivariate analysis of predictors of adiponectin levels and the explanation for the missingness is not provided.

4) There appears to be a missed opportunity to analyze or comment on the prevalence of SNVs associated with adiponectin levels in the centenarians vs. the very old or general background population. Studying centenarians as a phenotype is challenging because one cannot discern whether observed phenotypes are protective ("What got them to centenarian age"), adaptive or maladaptive with regards to aging-related physiologic changes, or an end-of-life phenotype. One exception to this comment is genetic variants that are present in centenarians with different frequencies compared to usual survival populations which can give us an idea about potentially protective (if enriched in centenarians) or detrimental variants (if not present in centenarians) for longevity. It would be interesting to know how the prevalence of identified genetic variants associated with adiponectin in centenarians compares to usual survival populations' prevalence as this could hint if adiponectin is protective or detrimental for exceptional longevity.

5) Some of the key covariates may need redefining. For instance, the paper describes that diabetes mellitus was defined by the level of A1c, without mention of the use of diabetes medications (a person with A1c of <6.5% might still have diabetes which is treated). BMI categorization should be reconsidered as a metabolic risk in Asian populations that occurs at lower levels of BMI, which is why many researchers and clinicians define overweight and obesity at lower BMI levels in Asian individuals.

6) What was the significance of taking the age range 85-89 as very old in this study taking in context with lifespan in the Japanese population? Please clarify. Why was it restricted to a 5-year time period rather than 80-89? Was it something to do with the available cohort?

7) What was the distribution of adiponectin levels in the cohort? Were their outliers present? How were outlier adiponectin levels handled in this study? What was the strategy implemented? Did it have an impact on the study?

8) Please mention the method used in the estimation of Adiponectin in the manuscript as it is the main component. Though from reference it's known that it is estimated by ELISA. Mention that in the manuscript itself as well as standardization and quality checks carried out.

9) Is gender distribution in this study representing population structure? How is the gender distribution of centenarians as well as the very old range in Japan?

10) Have the authors checked the power of the analysis and as well as checked the assumptions of the Cox model? Is the low number of participants in the subset analysis affecting the results? Do highlight in the limitation sections.

11) Authors talk about frailty in the discussion. Has frailty been defined in these cohorts? If so what was the association of adiponectin with frailty in different groups? Grip strength which is discussed is also an important component of physical frailty definition.

*Reviewer #1 (Recommendations for the authors):*

Previous studies have shown that, while adiponectin is associated with a favorable metabolic profile in the general population (lower BMI, better insulin sensitivity, and healthier lipid profiles), in older people it confers greater risk for mortality, the phenomenon that is sometimes called "adiponectin paradox" (PMID: 29438496). In this study, Sasaki and colleagues aimed to investigate factors associated with circulating high-molecular weight (cHMW) adiponectin levels and the associations between cHMW levels and mortality, among n=1,498 very old (age 85-89 years, 51.7% women) and n=812 centenarians (age >= 100, 84.4% women) from several study cohorts in Japan. This study design is primarily cross-sectional, with longitudinal measurements of adiponectin in a subset of very old participants and follow-up for mortality in another subset of participants. The study confirmed previously published associations between single nucleotide variants (SNV) in genes CDH13 (T-cadherin) and ADIPOQ (adiponectin gene) with adiponectin levels, and that adiponectin levels increase with age. The study also identified that metabolic factors traditionally associated with greater cHMW adiponectin (higher HDL, lower BMI) while associated with adiponectin in very old, have weaker associations with adiponectin in centenarians, especially centenarian men. They also conclude that high levels of cHMW adiponectin in very old men with low BMI are predictive of mortality, while adiponectin was not a significant mortality predictor in very old women or centenarian men or women.

The study provides useful data for the field of metabolic aspects of aging and longevity but it could be strengthened and extended by addressing the noted weaknesses.

Strengths:

– Large cohort of very old and centenarian individuals;

– Longitudinal measurements of adiponectin levels in a subset of participants confirming the observed cross-sectional age trend showing that adiponectin levels continue to increase into a very old age (included data until the age of 90);

– A wide range of available covariates;

– Sex-stratified analyses which address sex as a biological variable;

– Insightful comments about different physiologic meanings of BMI/adiposity in very old and centenarians (higher BMI protective) vs. younger individuals (higher BMI harmful).

Weaknesses:

– Conclusions about survival are not supported by data. Only one adjusted survival analysis is presented (very old men) and in that analysis adiponectin levels were not significantly associated with mortality after adjustment for BMI and other confounders. Other survival analyses are presented as unadjusted KM curves with log-rank tests, frequently with small numbers of participants and with crossing survival curves (questioning the assumption of proportional hazards). One potentially interesting finding is a possible greater mortality in very old men with low BMI and high adiponectin, compared to other combinations of BMI and adiponectin tertiles, including low BMI and mid and low adiponectin, which suggests that it is possible that some of the association between adiponectin and mortality is independent of the BMI at that time point (literature suggests that mortality association of adiponectin is abolished when adjusted for weight loss and low muscle mass in older adults PMID: 29438496).

– There is a substantial unexplained missingness of data. More than two-thirds of the participants in the very old group are missing mortality data and the reason for this was not provided. More than one-half of centenarians are not included in multivariate analysis of predictors of adiponectin levels and the explanation for the missingness is not provided.

– There appears to be a missed opportunity to analyze or comment on the prevalence of SNVs associated with adiponectin levels in the centenarians vs. very old or general background population. Studying centenarians as a phenotype is challenging because one cannot discern whether observed phenotypes are protective ("What got them to centenarian age"), adaptive or maladaptive with regards to aging-related physiologic changes, or an end-of-life phenotype. One exception to this comment is genetic variants that are present in centenarians with different frequencies compared to usual survival populations which can give us an idea about potentially protective (if enriched in centenarians) or detrimental variants (if not present in centenarians) for longevity. It would be interesting to know how the prevalence of identified genetic variants associated with adiponectin in centenarians compares to usual survival populations' prevalence as this could hint if adiponectin is protective or detrimental for exceptional longevity.

– Some of the key covariates may need redefining. For instance, the paper describes that diabetes mellitus was defined by the level of A1c, without mention of the use of diabetes medications (a person with A1c of <6.5% might still have diabetes which is treated). BMI categorization should be reconsidered as a metabolic risk in Asian populations that occurs at lower levels of BMI, which is why many researchers and clinicians define overweight and obesity at lower BMI levels in Asian individuals.

Abstract:

– Consider introducing in the background paragraph that SNVs for adiponectin will be assessed as genetic analysis comes in as a surprise in the next paragraph.

– Suggest including % male/female sex in the method paragraph.

– The statement about survival in the Results section is not supported by data. – Instead of lipid content suggest using lipid levels.

Introduction:

– When asserting that previous studies revealed that cHMW adiponectin enhanced sensitivity to insulin and other metabolic functions (lines 65-68) please specify if these were true causal relationships (presumably obtained with experiments in model organisms); also, the term "flexibility of adipose" is not clear.

– In the sentence that adiponectin knock-out mice were "viable" did the authors mean to say metabolically healthy or non-obese or similar (viable is interpreted as alive) (line 69).

– Consider elaborating more on the statement that "several reports revealed the physiologic significance of adiponectin in aging" (line 85).

– The statement "to elucidate its physiological function and significance in the oldest old" is an overstatement as this could not be done with this type of observational data.

Results:

– Supplementary Table 1 with baseline characteristics, as well as other Supplementary tables referenced in the manuscript were not provided.

– On the other hand, animal data in Supplementary Figures 9 and 10 come as a surprise in the Discussion as this was not mentioned previously; If authors intended to include these analyses, the other parts of the manuscript (abstract, methods, results) should reflect that.

– GWAS results in the text: Please include beats, p values do not tell anything about the effect size or direction.

– Authors mention that the analysis between adiponectin and HbA1c was done, but these results are not provided (line 143).

– The statement that the association between adiponectin and prevalent diabetes indicates that "adiponectin is involved in the T2DM pathway of DM development, regardless of age" (lines 147-8) is a statement of causality that is not supported by cross-sectional observational data.

– The results for total and LDL cholesterol were not provided (149-150).

Discussion

– Please clarify what is the novelty – while associations of adiponectin with age are not novel, this study adds more evidence that this association extends into very old and exceptionally old age.

– Statements of associations between adiponectin and mortality (lines 233-235 and elsewhere in the paper) are not supported by data as they were not independent of the major confounder (BMI), this should be clarified.

– Please consider adding a paragraph on the limitations (and strengths) of this study.

– Authors mention levels of cystatin C but this was not reported.

– As mentioned above, analysis of animal adipose is mentioned in the discussion but not elsewhere (unclear if this was an intended part of the manuscript)

– While the statement that "cHMW adiponectin could be considered a biomarker inversely associated with BMI" is correct, it is not comprehensive, given that high adiponectin is not only a marker of low adiposity but may also be a marker of low muscle mass and weight loss (features of frailty which dramatically increases mortality in older people).

Methods:

– Please elaborate more on the determination of DM status; was anything other than A1c considered (as noted in Public review), and was there any information about the type of DM?

Figures/tables:

– Table 1: consider presenting Cox PH results for other subgroups as well (it would be interesting to know what factors predict survival in centenarians in particular)

– Figure 1: consider removing cartoons depicting study participants as very frail; many very old or even centenarian people are in good health and functional status

– Figure 2 a-c: please clarify if these panels include data from all groups (very old, centenarian, men, women); for those not familiar with GWAS and other genetic analyses consider explaining briefly which alleles are a reference and which alternative.

– Figure 4: color-coding of variables (red, blue, green, orange) is not clear; also, please provide explanations of the abbreviations.

Supplementary tables are missing, including Suppl table 1 which is supposed to include baseline characteristics (with numbers of individuals with missing data hopefully).

– All figures with BMI categories: suggest renaming lowest BMI category to "underweight" as lean may be interpreted as normal BMI; also, BMI over 25 is not considered "obesity".

General:

– Throughout the paper and tables/ graphs: suggest using "very old" instead of "very older".

– Avoid implying causality from cross-sectional associations (another example in line 186).

– Consider including a flowchart that tracks which participants were included in which analyses (n seems to vary quite a bit).

*Reviewer #2 (Recommendations for the authors):*

High levels of adiponectin are found to be associated with mortality though most of these studies were carried out in age groups less than 80. In this paper, authors take a step forward to understand the distribution of adiponectin distribution in very old (85-89 years old) and centenarians. The authors have successfully presented valuable insights into this distribution as well as looked at its impact on mortality influenced by other physiological factors. Interestingly, authors have also investigated genetic variants associated with adiponectin levels in these groups using genome-wide approaches. This has led to the successful identification of genetic variants in CDH13 and ADIPOQ to be associated with adiponectin expression. In very older men, having low BMI and higher adiponectin levels were found to be a risk for mortality but not for very older women as well as in centenarians.

The positive part of this study is that it provides a picture of adiponectin distribution in these age groups and finds its implications which can be used as a reference study in the field. One concern might be the low sample size in the subgroup analysis where participants were stratified based on adiponectin level and BMI which has to be discussed in the limitation of the study. Another interesting point to look forward to is the impact of age itself in driving outcomes in this study including mortality. In centenarians, more than 99% of participants died in course of the study meaning more than adiponectin, it was complex age-associated multifactorial changes happenings which might be regulating mortality. It would be also interesting to look at the adiponectin levels in the missing age group of 90-100 years in the future.

The manuscript presents a detailed description of cHMW adiponectin levels in the very old and centenarians. Authors have looked at the impact of other factors influencing adiponectin levels including BMI as well as even ran a genome-wide analysis finding significant variants associated with adiponectin expression. Overall this manuscript is well-planned and provides a comprehensive overview of adiponectin and its influences on the elder population. There are some clarifications required.

1) Low incidence of Diabetes was reported in the centenarians. Was it because participants with diabetes were filtered out of the population with age? Will that have an effect on the adiponectin level association with diabetes with age?

2) Participants who were presented with ages between 89-100 were excluded from the analysis. Why are they represented in Figure 1b?

3) High level of Adiponectin was not associated with mortality in centenarians. Is that mainly because age itself is a risk to death or time to death is very low for adiponectin to have an effect? Also, 99% of participants succumbed to death in course of the study.

4) What was the mean or median follow-up time in Cox proportional hazard model in various groups?

5) Was GWAS adjusted for age or just gender? In the methods, its states model was adjusted with gender What other variables were included in the model? If not adjusted with age, why?

6) The writing has been acceptable throughout the manuscript except for the Abstract. I would like the authors to make the abstract more precise.

7) Based on adiponectin level grouping has been done into three groups which had again grouped into multiple groups based on BMI. This might be taking the power of the analysis. Is it possible to have two group classifications like "high" level of adiponectin against "rest" or the reverse and then subgrouping based on BMI?

8) It would be important to highlight a lower number of participants in subgroup analysis. Maybe the lower number of participants influencing the results and maybe it won't be perfect to come up with a precise conclusion. Please highlight these issues in the discussion.

9) Though the recruitment start date might have been referenced in the references mentioned. It would to helpful for readers to have when the study recruitment was started, and the years that followed in the text of the manuscript itself.

"In the very old, 100 (43.1%) men and 95 (32.2%) women died after six years; in centenarians, 122 (99.2%) men and 660 (99.5%) women died until 2016."

Please refer clearly to the start and end points of the analysis.

---

## [Author Response]

Essential revisions:(1) Conclusions about survival are not supported by data. Only one adjusted survival analysis is presented (very old men) and in that analysis adiponectin levels were not significantly associated with mortality after adjustment for BMI and other confounders. Other survival analyses are presented as unadjusted KM curves with log-rank tests, frequently with small numbers of participants and with crossing survival curves (questioning the assumption of proportional hazards).

As noted by the editor and reviewers, several of our previous results included conclusions based on unadjusted survival analyses. The KAWP cohort did not have survival information for the very old. However, we recently had access to the survival information, so we re-analyzed the survival analysis against the very old using the Cox proportional hazards method. We have also added information for the proportional hazards assumption (Supplementary Figures 7-8). In addition, it was not possible to obtain sufficient number of samples and events for centenarian men and cause-specific mortality; this has been stated as a limitation in the Discussion section.

The text was revised as follows;

“Higher cHMW adiponectin levels in very old men was positively associated with high all-cause mortality rates, independent of BMI.

High cHMW adiponectin levels are associated with increased all-cause mortality of CVD risk in adults in their 60s and 70s^12, 13, 14, 15^. To evaluate the effects of cHMW adiponectin levels on mortality in the very old and centenarians, hazard ratios of all-cause mortality were analyzed using Cox promotional hazards models for three quantiles of cHMW adiponectin levels (i.e., high, intermediate, and low) in 1,425 very old (678 men and 747 women) and 545 centenarians (90 men and 455 women) for whom both survival time information and a number of covariates were available. Prior to the analysis, the availability of sufficient samples and events for all-cause mortality were ensured for the survival analysis of the very old and centenarian women and there was no significant difference in the proportional hazards assumption of the cHMW adiponectin level and each of the covariates (Supplementary Figures 7-8). However, statistical power analysis indicated that there were insufficient events and the samples were ensured for the centenarian men even when they were divided into two groups. Within the follow-up periods, 145 (21.3%) men and 101 (13.5%) women died in the very old, whereas 89 (98.9%) men and 542 (99.4%) women died in the centenarians (Figure 5 and Supplementary File 1). As a result, the hazard ratios of all-cause mortality for intermediate and high levels of cHMW adiponectin groups in very old men were significantly higher (HR: 1.67 and 2.32) rather than those of the all-cause mortality of the low cHMW adiponectin level group (reference), even after adjustment for BMI (HR: 1.60 and 2.12). In contrast, the hazard ratio for all-cause mortality was significantly higher for the high cHMW adiponectin levels group in very old women was significantly higher (HR: 1.89), but was not significant after adjustment for BMI (HR: 1.41, (Figure 5)). This trend was also observed in the centenarian women.

To further elucidate the factors associated with mortality, we also analyzed cause-specific mortality associated with cancer, CVD, and pneumonia in the very old (Figure 5—figure supplements 3–5). The total number of events for each cause-specific mortality was 59 (cancer), 53 (CVD), and 40 (pneumonia), indicating that the analysis lacked sufficient statistical power. Testing populations with a 5% difference in event frequency would require approximately 440 samples for each group.”

“(3) Cox regression for all-cause mortality in centenarian men and cause-specific mortality in the very old men was statistically underpowered due to the insufficient size of samples and/or events. CVD mortality in very old men showed a trend to be associated with cHMW adiponectin levels, but statistically, twice the number of events or twice the number of total samples are needed to assess this.”

(2) One potentially interesting finding is a possible greater mortality in very old men with low BMI and high adiponectin, compared to other combinations of BMI and adiponectin tertiles, including low BMI and mid and low adiponectin, which suggests that it is possible that some of the association between adiponectin and mortality is independent of the BMI at that time point (literature suggests that mortality association of adiponectin is abolished when adjusted for weight loss and low muscle mass in older adults PMID: 29438496).

We thank the reviewers for their careful reading. Previous results of the mortality analysis for the very old were due to the unavailability of mortality data for half of the very old; therefore, we used about half the number of samples. Currently, mortality data are available for the KAWP cohort. We have reanalyzed the results using Cox regression with very old 678 men and 747 women with BMI as a continuous variable. As a result, adiponectin concentration was associated with all-cause mortality independently of BMI (Figure 5).

The text was revised according to question (1).

(3) There is a substantial unexplained missingness of data. More than two-thirds of the participants in the very old group are missing mortality data and the reason for this was not provided. More than one-half of centenarians are not included in multivariate analysis of predictors of adiponectin levels and the explanation for the missingness is not provided.

We apologize if the data conveyed such a remark. At the time of the first submission, the sample size for the mortality analysis was 2/3, due to the unavailability of mortality data for the very old KAWP cohort; these data are now available. Therefore, we have reanalyzed the data for the mortality analysis.

Among the centenarians, there were more missing values due to limitations related to their condition at the time of the survey. Therefore, the analysis was performed on 352 samples with no missing values for 26 variables used in the LASSO multiple regression analysis. As these situations are not comprehensible, a cohort overview and a summary of the number of samples in each analysis have been added as the Supplementary Figures1 and 2. Analyses with many missing values were described as a limitation of the study.

The text was revised as follows;

“This study had the following limitations: (1) Surveys of centenarians tend to have many missing values due to their limited physical and cognitive function; therefore, multivariate analysis using a series of covariates tends to reduce the number of samples to be analyzed.”

(4) There appears to be a missed opportunity to analyze or comment on the prevalence of SNVs associated with adiponectin levels in the centenarians vs. the very old or general background population. Studying centenarians as a phenotype is challenging because one cannot discern whether observed phenotypes are protective ("What got them to centenarian age"), adaptive or maladaptive with regards to aging-related physiologic changes, or an end-of-life phenotype. One exception to this comment is genetic variants that are present in centenarians with different frequencies compared to usual survival populations which can give us an idea about potentially protective (if enriched in centenarians) or detrimental variants (if not present in centenarians) for longevity. It would be interesting to know how the prevalence of identified genetic variants associated with adiponectin in centenarians compares to usual survival populations' prevalence as this could hint if adiponectin is protective or detrimental for exceptional longevity.

We thank you for this constructive suggestion. The allele frequency of cHMW-associated SNVs should have been compared among the Japanese population and our cohorts. Therefore, we have added a comparison of the minor allele frequency (MAF) of SNVs in Japanese controls (database), the very old, and centenarians in Supplementary Figure 5. As a result, there was no systematic MAF difference using Fisher’s exact test and multiple testing, although MAF in ADIPOQ was 3.1% higher in centenarian women. In addition, reviewer2 (5) commented that age should be considered as an adjustment variable in the GWAS, which we completely agree with. Therefore, the GWAS was re-analyzed with age as an adjustment variable.

The text has been revised as follows;

“To confirm the association between these SNVs and cHMW adiponectin levels, we determined the genotype of these two SNVs against the very old and centenarians using a TaqMan assay. As a result, no significant difference in minor allele frequency was found between Japanese control (ToMMo 38KJPN), the very old, and centenarian men and women using Fisher’s exact test and multiple testing (Figure 2—figure supplement 2).”

(5) Some of the key covariates may need redefining. For instance, the paper describes that diabetes mellitus was defined by the level of A1c, without mention of the use of diabetes medications (a person with A1c of <6.5% might still have diabetes which is treated). BMI categorization should be reconsidered as a metabolic risk in Asian populations that occurs at lower levels of BMI, which is why many researchers and clinicians define overweight and obesity at lower BMI levels in Asian individuals.

We thank the reviewer for the valuable suggestion. In line with this, we have redefined the criteria for diabetes mellitus as an individual meeting one or more of the following criteria; (1) HbA1c ≥ 6.5, (2) receiving antidiabetic drug therapy, and (3) receiving insulin injections. Because categorization by the combination of BMI and cHMW adiponectin level is difficult to ensure sufficient statistical power, BMI was entered into the Cox regression as a continuous variable.

The text was revised as follows;

“A person with DM was defined as follows: individuals with glycated hemoglobin (HbA1c) ≥ 6.5%, those receiving antidiabetic drug therapy, or those receiving insulin injections.”

(6) What was the significance of taking the age range 85-89 as very old in this study taking in context with lifespan in the Japanese population? Please clarify. Why was it restricted to a 5-year time period rather than 80-89? Was it something to do with the available cohort?

At baseline, participants in both the TOOTH and KAWP cohorts were over 85 years of age. Therefore, we set the age range for this study at 85–89 years.

(7) What was the distribution of adiponectin levels in the cohort? Were their outliers present? How were outlier adiponectin levels handled in this study? What was the strategy implemented? Did it have an impact on the study?

All data for cHMW adiponectin levels are shown in Figure 1; there are no outliers for cHMW adiponectin levels in this study.

(8) Please mention the method used in the estimation of Adiponectin in the manuscript as it is the main component. Though from reference it's known that it is estimated by ELISA. Mention that in the manuscript itself as well as standardization and quality checks carried out.

We apologize for the lack of information. We have added the method for measuring cHMW adiponectin levels by ELISA in the Methods section.

The text was added as follows;

“Measurement of cHMW adiponectin levels

The plasma cHMW adiponectin levels were measured using the Human HMW Adiponectin/ Acrp30 Immunoassay Quantikine ELISA Kit (R&D Systems, Inc, Minneapolis, MN, USA) according to manufacture protocol.”

(9) Is gender distribution in this study representing population structure? How is the gender distribution of centenarians as well as the very old range in Japan?

According to the 2020 Japanese Census, the proportion of women in the 85–89 age group was 0.644, and the proportion of centenarians was 0.877. This information has been added in the text as follows.

“The data for cHMW adiponectin levels were available for 812 centenarians (woman: 84.4%, 87.7% in Japanese census data in 2020) and 1,498 very old (woman: 51.7%, 64.4% in Japanese census data in 2020, Supplementary Figure 1). Participant characteristics at enrollment are presented in Supplementary Table 1. The flow chart for the analysis is shown in Figure 1—figure supplement 2.”

(10) Have the authors checked the power of the analysis and as well as checked the assumptions of the Cox model? Is the low number of participants in the subset analysis affecting the results? Do highlight in the limitation sections.

We performed a power analysis and confirmed that the all-cause mortality analysis for the very old and centenarian women included sufficient numbers of samples and events. However, centenarian men and cause-specific mortality in the very old population did not have sufficient numbers of samples and events. These are described in the results and mentioned as a limitation in the Discussion section.

“However, the statistical power analysis indicated that there were not sufficient events, and samples were ensured for the centenarian men even if they were divided into two groups.”

“To further elucidate the factors associated with mortality, we also analyzed cause-specific mortality associated with cancer, CVD, and pneumonia in the very old (Figure 5—figure supplements 3–5). The total number of events for each cause-specific mortality was 59 (cancer), 53 (CVD), and 40 (pneumonia), indicating that the analysis lacked sufficient statistical power. Testing populations with a 5% difference in event frequency would require approximately 440 samples for each group.”

“(3) Cox regression for all-cause mortality in centenarian men and cause-specific mortality in the very old men was statistically underpowered due to the insufficient size of samples and/or events. CVD mortality in very old men showed a trend to be associated with cHMW adiponectin levels, but statistically, twice the number of events or twice the number of total samples are needed to assess this.”

(11) Authors talk about frailty in the discussion. Has frailty been defined in these cohorts? If so what was the association of adiponectin with frailty in different groups? Grip strength which is discussed is also an important component of physical frailty definition.

We appreciate this insightful comment. This is a very important point and there are few previous reports on this factor. However, frailty analysis for centenarians is difficult because most centenarians would be classified as frail using the current frailty criteria. For the very old, only the KAWP, one of the very old cohorts, collected sufficient data to calculate frailty.

Therefore, results from previous publications and analyses of selected cohort data in this study have been added to the Discussion section as follows;

“Frailty is an important concept in health maintenance and the process of functional decline in the oldest old. Recently, serum adiponectin levels have been positively associated with frailty in the oldest old34, 35. In our cohort, most centenarians were classified as frail according to the current frailty criteria, so it is difficult to assess frailty in centenarians. For the very old, only the KAWP, one of the cohorts that included the very old, collected sufficient data to assess frailty. Using these limited data for the very old, we analyzed the distribution of cHMW adiponectin levels in each frailty category and analyzed their association with the revised J-CHS frailty index criteria using multiple regression analysis36. As a result, we found that cHMW adiponectin levels were significantly associated with frailty, both in very old men and women (Supplementary File 12). The cHMW adiponectin level was also significantly associated with frailty in very old women even after adjustment for BMI; however, no significant association was observed in very old men after adjustment by BMI. Thus, cHMW adiponectin levels would be associated with frailty in the very old, especially in women.”

“(4) The analysis of cHMW adiponectin levels and frailty in centenarians is difficult because most centenarians would be classified as frail according to the current frailty criteria. Of the two cohort studies of very old participants, the TOOTH study did not have sufficient data adjusted for the evaluation of J-CHS frailty criteria. Therefore, the association between cHMW adiponectin levels and frailty was analyzed in selected samples derived only from the KAWP study only. This was only a cross-sectional analysis, and further analysis would be needed to prove causality. Therefore, these are described only in the Discussion section.”

Reviewer #1 (Recommendations for the authors):Previous studies have shown that, while adiponectin is associated with a favorable metabolic profile in the general population (lower BMI, better insulin sensitivity, and healthier lipid profiles), in older people it confers greater risk for mortality, the phenomenon that is sometimes called "adiponectin paradox" (PMID: 29438496). In this study, Sasaki and colleagues aimed to investigate factors associated with circulating high-molecular weight (cHMW) adiponectin levels and the associations between cHMW levels and mortality, among n=1,498 very old (age 85-89 years, 51.7% women) and n=812 centenarians (age >= 100, 84.4% women) from several study cohorts in Japan. This study design is primarily cross-sectional, with longitudinal measurements of adiponectin in a subset of very old participants and follow-up for mortality in another subset of participants. The study confirmed previously published associations between single nucleotide variants (SNV) in genes CDH13 (T-cadherin) and ADIPOQ (adiponectin gene) with adiponectin levels, and that adiponectin levels increase with age. The study also identified that metabolic factors traditionally associated with greater cHMW adiponectin (higher HDL, lower BMI) while associated with adiponectin in very old, have weaker associations with adiponectin in centenarians, especially centenarian men. They also conclude that high levels of cHMW adiponectin in very old men with low BMI are predictive of mortality, while adiponectin was not a significant mortality predictor in very old women or centenarian men or women.The study provides useful data for the field of metabolic aspects of aging and longevity but it could be strengthened and extended by addressing the noted weaknesses.Strengths:– Large cohort of very old and centenarian individuals;– Longitudinal measurements of adiponectin levels in a subset of participants confirming the observed cross-sectional age trend showing that adiponectin levels continue to increase into a very old age (included data until the age of 90);– A wide range of available covariates;– Sex-stratified analyses which address sex as a biological variable;– Insightful comments about different physiologic meanings of BMI/adiposity in very old and centenarians (higher BMI protective) vs. younger individuals (higher BMI harmful).Weaknesses:– Conclusions about survival are not supported by data. Only one adjusted survival analysis is presented (very old men) and in that analysis adiponectin levels were not significantly associated with mortality after adjustment for BMI and other confounders. Other survival analyses are presented as unadjusted KM curves with log-rank tests, frequently with small numbers of participants and with crossing survival curves (questioning the assumption of proportional hazards). One potentially interesting finding is a possible greater mortality in very old men with low BMI and high adiponectin, compared to other combinations of BMI and adiponectin tertiles, including low BMI and mid and low adiponectin, which suggests that it is possible that some of the association between adiponectin and mortality is independent of the BMI at that time point (literature suggests that mortality association of adiponectin is abolished when adjusted for weight loss and low muscle mass in older adults PMID: 29438496).– There is a substantial unexplained missingness of data. More than two-thirds of the participants in the very old group are missing mortality data and the reason for this was not provided. More than one-half of centenarians are not included in multivariate analysis of predictors of adiponectin levels and the explanation for the missingness is not provided.– There appears to be a missed opportunity to analyze or comment on the prevalence of SNVs associated with adiponectin levels in the centenarians vs. very old or general background population. Studying centenarians as a phenotype is challenging because one cannot discern whether observed phenotypes are protective ("What got them to centenarian age"), adaptive or maladaptive with regards to aging-related physiologic changes, or an end-of-life phenotype. One exception to this comment is genetic variants that are present in centenarians with different frequencies compared to usual survival populations which can give us an idea about potentially protective (if enriched in centenarians) or detrimental variants (if not present in centenarians) for longevity. It would be interesting to know how the prevalence of identified genetic variants associated with adiponectin in centenarians compares to usual survival populations' prevalence as this could hint if adiponectin is protective or detrimental for exceptional longevity.– Some of the key covariates may need redefining. For instance, the paper describes that diabetes mellitus was defined by the level of A1c, without mention of the use of diabetes medications (a person with A1c of <6.5% might still have diabetes which is treated). BMI categorization should be reconsidered as a metabolic risk in Asian populations that occurs at lower levels of BMI, which is why many researchers and clinicians define overweight and obesity at lower BMI levels in Asian individuals.Abstract:– Consider introducing in the background paragraph that SNVs for adiponectin will be assessed as genetic analysis comes in as a surprise in the next paragraph.– Suggest including % male/female sex in the method paragraph.

Thank you for your suggestions. We have added %male/female sex information in the Method paragraph as follows,

“Methods: The study included 812 (women: 84.4%) for centenarians and 1,498 (women: 51.7%) for the very old.”

– The statement about survival in the Results section is not supported by data. – Instead of lipid content suggest using lipid levels.Introduction:– When asserting that previous studies revealed that cHMW adiponectin enhanced sensitivity to insulin and other metabolic functions (lines 65-68) please specify if these were true causal relationships (presumably obtained with experiments in model organisms); also, the term "flexibility of adipose" is not clear.

Thank you for your suggestions. We have changed the sentence as follows:

“Previous studies in the mouse model studies have shown that cHMW adiponectin enhances insulin sensitivity and plasma lipid clearance; high levels of cHMW adiponectin improve the stability of lipid homeostasis and provide systemic tolerance to obesity under normal physiological conditions^4, 5, 6^.”

– In the sentence that adiponectin knock-out mice were "viable" did the authors mean to say metabolically healthy or non-obese or similar (viable is interpreted as alive) (line 69).– Consider elaborating more on the statement that "several reports revealed the physiologic significance of adiponectin in aging" (line 85).– The statement "to elucidate its physiological function and significance in the oldest old" is an overstatement as this could not be done with this type of observational data.

Thank you for your suggestions. We agree with you. We have changed the sentence as follows:

“To provide evidence for understanding the physiological function and significance of adiponectin in the oldest old, this study aimed to determine the status and factors associated with cHMW adiponectin levels in 2,310 adults aged ≥85 years, including 812 centenarians.”

Results:– Supplementary Table 1 with baseline characteristics, as well as other Supplementary tables referenced in the manuscript were not provided.– On the other hand, animal data in Supplementary Figures 9 and 10 come as a surprise in the Discussion as this was not mentioned previously; If authors intended to include these analyses, the other parts of the manuscript (abstract, methods, results) should reflect that.– GWAS results in the text: Please include beats, p values do not tell anything about the effect size or direction.

Accordingly, we have added the Z-score information to the GWAS figure to show the effect size and direction in Figure 2.

– Authors mention that the analysis between adiponectin and HbA1c was done, but these results are not provided (line 143).

We monitored the criteria for DM using HbA1c level, anti-diabetic medication, and insulin injection information because HbA1c level information alone was not sufficient, as indicated by reviewer 1. Therefore, we have deleted this data.

– The statement that the association between adiponectin and prevalent diabetes indicates that "adiponectin is involved in the T2DM pathway of DM development, regardless of age" (lines 147-8) is a statement of causality that is not supported by cross-sectional observational data.

Thank you for pointing this out. We agree with you. We have revised the sentence as follows:

“We found that cHMW adiponectin levels in the DM group were significantly lower than those in the non-DM group in both the very old and centenarians, indicating that adiponectin is associated with the DM pathway, regardless of age.”

– The results for total and LDL cholesterol were not provided (149-150).

This has been included in Supplementary Table 1.

Discussion– Please clarify what is the novelty – while associations of adiponectin with age are not novel, this study adds more evidence that this association extends into very old and exceptionally old age.

Thank you for the opportunity to provide clarity here. We have revised the sentence as follows:

“The results of this study showed that cHMW adiponectin levels increased with age up to centenarians, although the associated factors varied with sex. Therefore, we are further elucidating whether the increment of cHMW adiponectin level with age extends to very old and exceptionally old age.”

– Statements of associations between adiponectin and mortality (lines 233-235 and elsewhere in the paper) are not supported by data as they were not independent of the major confounder (BMI), this should be clarified.

Thank you for your suggestions. We have added data on the association between death and BMI to Figure 5.

– Please consider adding a paragraph on the limitations (and strengths) of this study.

In light of your comment, we have added a paragraph summarizing the limitations of the study at the end of the Discussion section. The following has been inserted:

“The study had the following limitations: (1) Surveys of centenarian surveys tend to have many missing values due to their limited physical and cognitive function; therefore, multivariate analysis using a series of covariates tends to reduce the number of samples to be analyzed. (2) Although the short survival time of centenarian in this showed no association between cHMW adiponectin level and all-cause mortality in this study, strong factors associated with survival, such as N-terminal pro-brain natriuretic peptide (NTproBNP) and albumin (ALB), tend to be detectable, while weaker factors are more difficult to detect. (3) Cox regression for all-cause mortality in centenarian men and cause-specific mortality in the very old men was underpowered due to the statistically insufficient size of samples and/or events. CVD mortality in very old men showed a trend to be associated with cHMW adiponectin levels, but statistically, twice the number of events or twice the number of total samples are needed to assess this. (4) Analysis of cHMW adiponectin levels and frailty in centenarians is difficult because most centenarians would be classified as frail according to the current frailty criteria. Of the two cohort studies of very old participants, the TOOTH study did not have sufficient data adjusted for J-CHS frailty criteria. Therefore, the association between cHMW adiponectin levels and frailty was analyzed in selected samples from the KAWP study only. This was only a cross-sectional analysis, and further analysis would be needed to prove causality. Therefore, these are described only in the Discussion section.”

– Authors mention levels of cystatin C but this was not reported.

We have supplied cystatin C information in Supplementary Table 1.

– As mentioned above, analysis of animal adipose is mentioned in the discussion but not elsewhere (unclear if this was an intended part of the manuscript)

This analysis was based on the results of another group; therefore, it is not included in the Results section, but only mentioned in the Discussion section.

– While the statement that "cHMW adiponectin could be considered a biomarker inversely associated with BMI" is correct, it is not comprehensive, given that high adiponectin is not only a marker of low adiposity but may also be a marker of low muscle mass and weight loss (features of frailty which dramatically increases mortality in older people).

In view of your comment, we have deleted the discrepant sentence.

Methods:– Please elaborate more on the determination of DM status; was anything other than A1c considered (as noted in Public review), and was there any information about the type of DM?

Thank you for pointing this out. We have re-defined the criteria for diabetes mellitus and inserted this in the methods section as follows:

“A person with DM was defined as follows: individuals with glycated hemoglobin (HbA1c) ≥ 6.5%, those receiving antidiabetic drug therapy, or those receiving insulin injections (Figure 3a,b).”

Figures/tables:– Table 1: consider presenting Cox PH results for other subgroups as well (it would be interesting to know what factors predict survival in centenarians in particular)– Figure 1: consider removing cartoons depicting study participants as very frail; many very old or even centenarian people are in good health and functional status

Thank you for pointing out the inappropriateness of these cartoons. We have removed them accordingly.

– Figure 2 a-c: please clarify if these panels include data from all groups (very old, centenarian, men, women); for those not familiar with GWAS and other genetic analyses consider explaining briefly which alleles are a reference and which alternative.

Thank you for the opportunity to provide clarity here. We have added this information in the legend of Figure 2.

– Figure 4: color-coding of variables (red, blue, green, orange) is not clear; also, please provide explanations of the abbreviations.

We apologize for the lack of clarity here. The information was described in the Supplementary Figure, but not in Figure 4 in the previous version. We have now revised this same and added this information in Figure 4.

Supplementary tables are missing, including Suppl table 1 which is supposed to include baseline characteristics (with numbers of individuals with missing data hopefully).– All figures with BMI categories: suggest renaming lowest BMI category to "underweight" as lean may be interpreted as normal BMI; also, BMI over 25 is not considered "obesity".

Based on the reviewer's comments, we have considered BMI as a continuous variable in the revised version of the manuscript. As a result, we have removed the BMI category from the manuscript.

General:– Throughout the paper and tables/ graphs: suggest using "very old" instead of "very older".

We have revised the term throughout the manuscript based on your valuable suggestion.

– Avoid implying causality from cross-sectional associations (another example in line 186).

Thank you for pointing this out. We agree with you. The entire manuscript was reviewed and revised accordingly.

– Consider including a flowchart that tracks which participants were included in which analyses (n seems to vary quite a bit).

Thank you for your suggestion. We have added new Supplementary Figures 1-2, which include a flowchart describing which participants were included in which analyses.

Reviewer #2 (Recommendations for the authors):High levels of adiponectin are found to be associated with mortality though most of these studies were carried out in age groups less than 80. In this paper, authors take a step forward to understand the distribution of adiponectin distribution in very old (85-89 years old) and centenarians. The authors have successfully presented valuable insights into this distribution as well as looked at its impact on mortality influenced by other physiological factors. Interestingly, authors have also investigated genetic variants associated with adiponectin levels in these groups using genome-wide approaches. This has led to the successful identification of genetic variants in CDH13 and ADIPOQ to be associated with adiponectin expression. In very older men, having low BMI and higher adiponectin levels were found to be a risk for mortality but not for very older women as well as in centenarians.The positive part of this study is that it provides a picture of adiponectin distribution in these age groups and finds its implications which can be used as a reference study in the field. One concern might be the low sample size in the subgroup analysis where participants were stratified based on adiponectin level and BMI which has to be discussed in the limitation of the study. Another interesting point to look forward to is the impact of age itself in driving outcomes in this study including mortality. In centenarians, more than 99% of participants died in course of the study meaning more than adiponectin, it was complex age-associated multifactorial changes happenings which might be regulating mortality. It would be also interesting to look at the adiponectin levels in the missing age group of 90-100 years in the future.The manuscript presents a detailed description of cHMW adiponectin levels in the very old and centenarians. Authors have looked at the impact of other factors influencing adiponectin levels including BMI as well as even ran a genome-wide analysis finding significant variants associated with adiponectin expression. Overall this manuscript is well-planned and provides a comprehensive overview of adiponectin and its influences on the elder population. There are some clarifications required.(1) Low incidence of Diabetes was reported in the centenarians. Was it because participants with diabetes were filtered out of the population with age? Will that have an effect on the adiponectin level association with diabetes with age?

We thank the reviewer for the careful reading. As suggested, this possibility cannot be excluded, so we have added this to the discussion as follows:

“We have previously reported that a low incidence of T2DM is a characteristic of centenarians; therefore, we deduced that the high cHMW adiponectin levels in centenarians would be partially influenced by a low incidence of T2DM.”

(2) Participants who were presented with ages between 89-100 were excluded from the analysis. Why are they represented in Figure 1b?

We apologize for the oversight. We have removed the very old aged 90–99 years from Figure 1B.

(3) High level of Adiponectin was not associated with mortality in centenarians. Is that mainly because age itself is a risk to death or time to death is very low for adiponectin to have an effect? Also, 99% of participants succumbed to death in course of the study.

Thank you for the opportunity to provide clarity here. Even in centenarians, strong factors such as BMI and NTproBNP are associated with all-cause mortality. However, as the reviewer points out, this possibility cannot be excluded; therefore, we have listed these as limitations in the Discussion section as follows.

“(2) Although the short survival time of centenarians in this study showed no association between cHMW adiponectin level and all-cause mortality, strong factors associated with survival, such as N-terminal pro-brain natriuretic peptide (NTproBNP) and albumin (ALB), tend to be detectable, while weaker factors are more difficult to detect.”

(4) What was the mean or median follow-up time in Cox proportional hazard model in various groups?

Accordingly, we have added this information in Supplementary File 1.

(5) Was GWAS adjusted for age or just gender? In the methods, its states model was adjusted with gender What other variables were included in the model? If not adjusted with age, why?

Thank you for your pertinent query. We have re-analyzed all GWAS adjusted with both age and sex.

(6) The writing has been acceptable throughout the manuscript except for the Abstract. I would like the authors to make the abstract more precise.

We apologize if the data conveyed such a remark. We have re-written the abstract as follows, per your suggestion.

“Abstract

Background: High levels of circulating adiponectin are associated with increased insulin sensitivity, low prevalence of diabetes, and low body mass index (BMI); however, high levels of circulating adiponectin are also associated with increased mortality in the 60–70 age group. In this study, we aimed to clarify factors associated with circulating high-molecular-weight (cHMW) adiponectin levels and their association with mortality in the very old (85–89 years old) and centenarians.

Methods: The study included 812 (women: 84.4%) for centenarians and 1,498 (women: 51.7%) for the very old. The genomic DNA sequence data were obtained by whole genome sequencing or DNA microarray-imputation methods. LASSO and multivariate regression analyses were used to evaluate cHMW adiponectin characteristics and associated factors. All-cause mortality was analyzed in three quantile groups of cHMW adiponectin levels using Cox regression.

Results: The cHMW adiponectin levels were increased significantly beyond 100 years of age, were negatively associated with diabetes prevalence, and were associated with SNVs in CDH13 (p = 2.21 × 10^-22^) and ADIPOQ (p = 5.72 × 10^-7^). Multivariate regression analysis revealed that genetic variants, BMI, and high-density lipoprotein cholesterol (HDLC) were the main factors associated with cHMW adiponectin levels in the very old, whereas the BMI showed no association in centenarians. The hazard ratios for all-cause mortality in the intermediate and high cHMW adiponectin groups in very old men were significantly higher rather than those for all-cause mortality in the low level cHMW adiponectin group, even after adjustment with BMI. In contrast, the hazard ratios for all-cause mortality were significantly higher for high cHMW adiponectin groups in very old women, but were not significant after adjustment with BMI.

Conclusions: cHMW adiponectin levels increased with age until centenarians, and the contribution of known major factors associated with cHMW adiponectin levels, including BMI and HDLC, varies with age, suggesting that its physiological significance also varies with age in the oldest old.”

(7) Based on adiponectin level grouping has been done into three groups which had again grouped into multiple groups based on BMI. This might be taking the power of the analysis. Is it possible to have two group classifications like "high" level of adiponectin against "rest" or the reverse and then subgrouping based on BMI?(8) It would be important to highlight a lower number of participants in subgroup analysis. Maybe the lower number of participants influencing the results and maybe it won't be perfect to come up with a precise conclusion. Please highlight these issues in the discussion.

Thank you for pointing this out. We completely agree with you. We have described these points as limitations in the Discussion section as follows,

“(3) Cox regression for all-cause mortality in centenarian men and cause-specific mortality in very old men was underpowered due to the statistically insufficient size of samples and/or events. CVD mortality in very old men showed a trend to be associated with cHMW adiponectin levels, but statistically, twice the number of events or twice the number of total samples are needed to assess this.”

(9) Though the recruitment start date might have been referenced in the references mentioned. It would to helpful for readers to have when the study recruitment was started, and the years that followed in the text of the manuscript itself."In the very old, 100 (43.1%) men and 95 (32.2%) women died after six years; in centenarians, 122 (99.2%) men and 660 (99.5%) women died until 2016."Please refer clearly to the start and end points of the analysis.

Thank you for the opportunity to provide clarity here. We have added described the cohort information in an additional Supplementary Figure1.